# Glacial isostatic adjustment reduces past and future Arctic subsea permafrost

Roger C. Creel [1,2], Frederieke Miesner [3] ✉, Stiig Wilkenskjeld [4], Jacqueline Austermann[1] & Pier Paul Overduin [3]

Sea-level rise submerges terrestrial permafrost in the Arctic, turning it into subsea permafrost. Subsea permafrost underlies ~1.8 million km² of Arctic continental shelf, with thicknesses in places exceeding 700 m. Sea-level variations over glacial-interglacial cycles control subsea permafrost distribution and thickness, yet no permafrost model has accounted for glacial isostatic adjustment (GIA), which deviates local sea level from the global mean due to changes in ice and ocean loading. Here we incorporate GIA into a pan-Arctic model of subsea permafrost over the last 400,000 years. Including GIA significantly reduces present-day subsea permafrost thickness, chiefly because of hydro-isostatic effects as well as deformation related to Northern Hemisphere ice sheets. Additionally, we extend the simulation 1000 years into the future for emissions scenarios outlined in the Intergovernmental Panel on Climate Change's sixth assessment report. We find that subsea permafrost is preserved under a low emissions scenario but mostly disappears under a high emissions scenario.

Sea-level low stands during past glacial periods exposed the Arctic continental shelf to cold air temperatures that froze the ground, forming up to a kilometer of new permafrost[1]. Postglacial sea-level rise inundated much of this cryotic sediment, producing subsea permafrost, which began to thaw as oceanic heat and salt propagated downwards from the seafloor[2]. Permafrost is defined here as sediment above or below sea level that has a temperature at or below 0 °C for at least 2 years and may or may not contain ice. While present-day subsea permafrost thaws due to geothermal heat from below and ocean warming from above, more is created at an accelerating rate as terrestrial permafrost turns into subsea permafrost through coastal erosion[3] and sea-level rise[4,5].

The need to track human carbon dioxide emissions (CO₂) has driven assessments of the global carbon budget, including the amount and stability of the carbon reservoir below the ocean floor[6,7]. The ongoing debate surrounding how much carbon from thawing subsea permafrost will reach the atmosphere[8–10] has precluded subsea permafrost's inclusion in global carbon budgets. Recent work and

structured expert assessment, however, suggest that the submarine permafrost domain may hold an amount of carbon in organic matter and methane hydrates of similar magnitude to the Earth's total gas reserves[8,11–13]. Rising Arctic water temperatures in the coming century, projected under all emissions scenarios, will hasten subsea permafrost thaw[14]. Accelerated permafrost thaw rates will increase carbon mobilization rates beneath the seabed. Since this carbon may reach the atmosphere as a greenhouse gas, it is important to have a more precise estimate of the amount of carbon currently trapped in and by permafrost, its stability, and the timing of its release.

Such an estimate requires accurate quantification of how much subsea permafrost exists today. Regional maps of present-day subsea permafrost extent typically rely on a combination of observations and physics-based modeling[15,16]. The International Permafrost Association (IPA) permafrost map, an early pan-Arctic effort, applied the heuristic that permafrost would exist anywhere where the shelf was exposed for long enough during sea-level lowstands to establish permafrost, implying unglaciated regions shallower than around 100 m[17]. More

[1]Lamont-Doherty Earth Observatory, Columbia University, New York, NY, USA. [2]Department of Physical Oceanography, Woods Hole Oceanographic Institution, Woods Hole, MA, USA. [3]Alfred Wegener Institute Helmholtz-Centre for Polar and Marine Research, Potsdam, Germany. [4]Max Planck Institute for Meteorology, Hamburg, Germany. ✉e-mail: fmiesner@awi.de

recently, subsea permafrost was mapped in a consistent manner at circum-Arctic spatial scale between 450 thousand years before present (kyr BP) and the present[18] by forcing a heat transfer model with spatially-varying geothermal heat flux, depth-varying ocean bottom water temperature, sediment porosity, global mean sea level (GMSL) from a Red Sea oxygen isotope record[19], and ice sheet thicknesses and air temperature from the CLIMBER2 Earth System Model[20].

Sea level and ice history are the most important controls on subsea permafrost formation. Together, they determine the fraction of time Arctic sediments are exposed to (relatively) warm temperatures beneath ice sheets or oceans rather than to cold air temperatures. In Arctic shelf regions beyond the maximal extents of the Northern Hemispheric ice sheets, inundation time controls the distribution, depth, and density of subsea permafrost[21]. Extant subsea permafrost calculations have included GMSL as a forcing term[2,18,22]. However, local sea level at locations on the Arctic shelf deviates from GMSL[23] due to glacial isostatic adjustment (GIA), which is the gravitational, deformational, and rotational response of the solid Earth to ice and liquid water loading[24]. In the GIA literature, local sea level is also often referred to as relative sea level (RSL), which is defined as sea level at a given location and time relative to present-day sea level at the same location.

The deviation between local and global mean sea levels is particularly pronounced near Banks Island and in the Barents and Kara Seas —where ice sheet loading deformsed the solid Earth by hundreds of meters over past glacial cycles−and along the western Laptev Sea and North Slope, which underwent peripheral bulge uplift and subsidence[25,26]. Even in places far from the Northern Hemisphere ice sheets at Last Glacial Maximum (LGM, ~26.5 to 19 kyr BP), such as the East Siberian Sea, changing water loading over glacial cycles can cause RSL to deviate from GMSL by 10+ meters[23]. Since these changes in local sea-level history can lengthen or shorten the duration of land inundation or seabed exposure for large portions of the Arctic shelf, we hypothesize that their omission leads to nonuniform biases in estimates of subsea permafrost distribution, thickness, and thaw rate.

Here we test this hypothesis by extending the subsea permafrost model of Overduin et al.[18] to include RSL produced by GIA modeling. We isolate the effects of GIA by comparing permafrost extents from a simulation that includes spatially varying RSL to two that do not. We then produce two additional simulations to test the model's sensitivity to GIA parameterization and ice history. We explore whether the inclusion of GIA in numerically modeled subsea permafrost improves correspondence between modeled and measured subsea permafrost extent. We further explore the effect of future warming scenarios on subsea permafrost distribution by extending models that do and do not include GIA to year 3000 under a range of ice melt scenarios related to Shared Socioeconomic Pathways (SSPs, hereafter 'emissions pathways') from the Intergovernmental Panel on Climate Change's 6th Assessment report [IPCC[27]].

## Results

Subsea permafrost distribution and state on the Arctic continental shelf was simulated from 400 kyr BP to the pre-industrial (1850 CE) using five model configurations: (1) the CLIMBER2 ice history[20] and GMSL curve from Grant et al.[19] without GIA (hereafter legacy run); (2) the ICE-6G ice history[28] and GMSL curve prior to the LGM from Waelbroeck et al.[29] without GIA (hereafter base run); and (3) the ICE-6G ice history and pre-LGM GMSL curve of Walbroeck et al.[29] with GIA using the VM5a viscosity structure (hereafter GIA run). Two additional model configurations were explored to investigate the model's sensitivity to GIA parameterization and ice history: (4) the ICE-6G ice history and pre-LGM GMSL curve of Waelbroeck et al.[29] with GIA using an alternative viscosity structure that resembles the 'high' viscosity solution of Lambeck et al.[30], hereafter the ANU solid Earth structure and GIA-2 run; and (5) a scenario identical to scenario 4 but with the ICE-6G northern

hemisphere ice sheets replaced with the ANU ice histories[25,30,31], hereafter the GIA-3 run (see the "Methods" section for further details).

The subsea permafrost calculation was extended from 1850 CE to 3000 CE for the GIA and base runs using 17 future ice sheet configurations based on the ISMIP6 ensemble[32,33] and climate forcing scenarios from the IPCC-AR6 (see the "Methods" section). The GIA run is presented hereafter, and we demonstrate and explain how changes in model setup between the legacy run, which resembles the model of Overduin et al.[18] (see the "Methods" section), the base run, the GIA run, and the two sensitivity tests GIA-2 and GIA-3 affect our modeling results.

Permafrost was modeled between 187 m below and 18 m above present-day sea level at every location on the Arctic continental shelf and nearshore. The total modeled permafrost area is defined as the sum of modeled regions whose depth profiles include terrestrial or subsea permafrost. Sedimentation rates, mineral conductivity, geothermal heat flux, and vertical conductive heat flux were parameterized following[18]. At every timestep in the resulting permafrost distribution, we removed permafrost from locations where warm bottom water from present-day rivers, deltas, and estuaries likely precludes permafrost formation[18].

### Past evolution and present-day extent

The temporal evolution of subsea permafrost, as measured by mean thickness, responds to Earth's sawtooth history of ice volume change (Fig. 1). The mean thickness of permafrost in the total model area increases during glaciations as sea level falls and exposes the shelf to cold air temperatures. Subsea permafrost is generally absent during these times since the continental shelves are exposed. Deglaciation inundates continental shelves and turns terrestrial permafrost into subsea permafrost, which quickly thaws as warm ocean waters increase temperatures on the shelf. After interglacials, subsea permafrost continues to thaw until it disappears or is converted to terrestrial permafrost by falling sea level. In the GIA run, the mean thickness of permafrost in our total modeled area peaks at 500−550 m during glacial maxima and thins to 125−150 m by the end of interglacials (Fig. 1B).

Based on the GIA run, subsea permafrost presently underlies 1.8 million square kilometers of the Arctic continental shelf and has a mean thickness of 253 m. Subsea permafrost reaches a maximum thickness of 708 m in shallow sediments offshore of Yukagir in the central Laptev Sea. Subsea permafrost that exceeds a thickness of 500 m also underlies the shallow central Kara Sea and the westernmost coastline of the Alaskan North Slope, while much of the deeper Chukchi and East Siberian Seas cover subsea permafrost that is <200 m thick (Fig. 2A).

### Ice history and global mean sea-level curve

The choice of ice history affects modeled present-day subsea permafrost. When compared to subsea permafrost estimates from the legacy run, adopting the base run results in thicker present-day cryotic sediment on the deep Russian continental shelf and nearly all of the Canadian Arctic by >50 m, but yields thinner cryotic sediment on much of the shallow Russian continental shelf by 50−150 m and in the eastern Kara Sea by >200 m (Fig. 2D).

These patterns are explained by the differing GMSL and ice distributions in the base and legacy runs. GMSL in the base run is generally higher early in glacial intervals (marine isotope stage (MIS) 11b−10b, 9d−8b, 7b−6b, 5d−3a) than GMSL in the legacy run, but lower during peak glacials (MIS 10a, 7d, 6a, 2, Fig. 3). This difference in GMSL has a pan-Arctic effect on subsea permafrost. Higher early-glacial GMSL inhibits the formation of shallow subsea permafrost everywhere in the Arctic by decreasing subaerial exposure time; lower peak-glacial GMSL enhances subsea permafrost formation on the deep shelf (Fig. 3). Subsea permafrost differences driven by ice sheet geometry

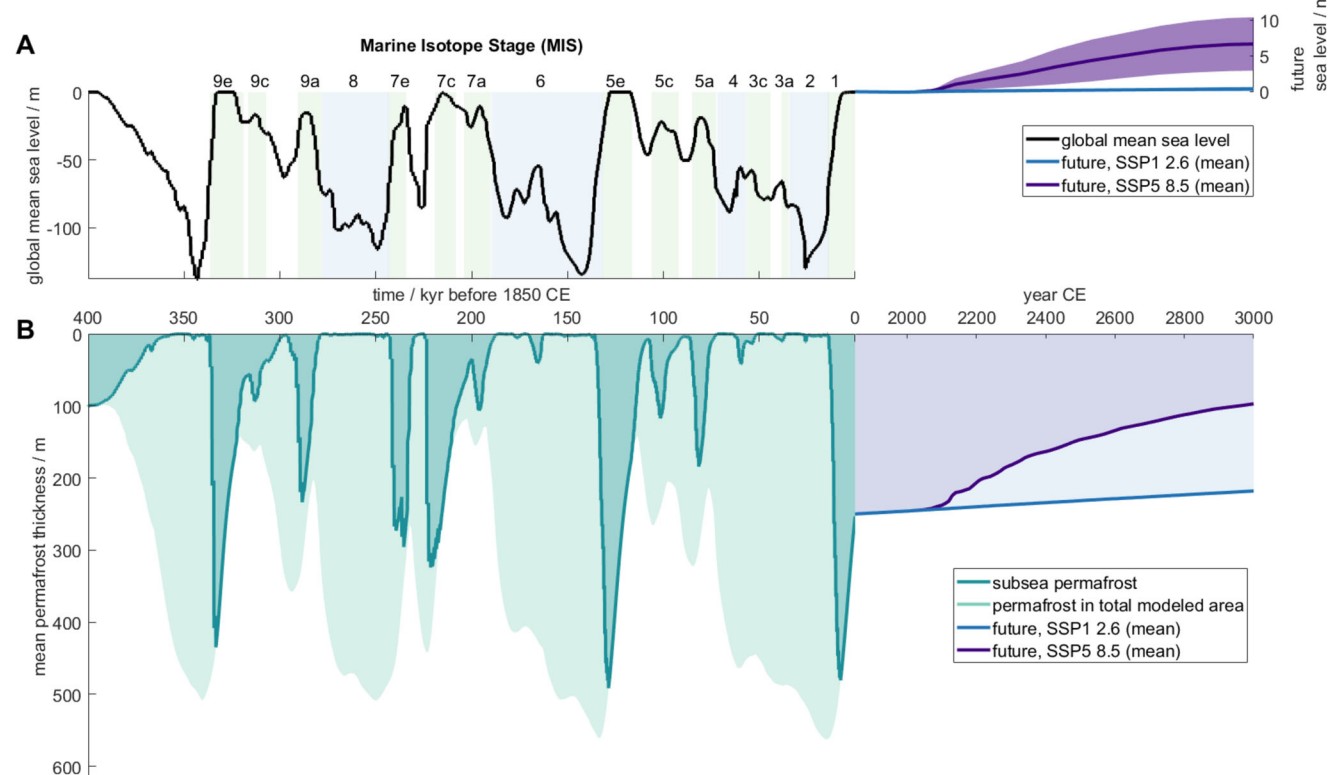

**Fig. 1 | Subsea permafrost thickness as a function of time from 400 kyr BP to 1850 CE and projected until 3000 CE.** Timeseries of global mean sea level (**A**) and subsea permafrost thickness (**B**). **A** Global mean sea level from Waelbroeck et al.[29] from 400 to 26 kyr BP, Peltier et al.[28] from 26 kyr BP to 1850 CE, (see the "Methods" section), and Greve, Chambers, et al.[32,35] for future projections, see the "Methods" section. Marine isotope stages are indicated following Railsback et al.[51]. **B** Mean subsea permafrost thickness (dark teal) between 400 kyr BP and 1850 CE for the GIA run. Mean permafrost thickness in the total modeled area (light teal). Mean permafrost thickness for low (SSP1–2.6, blue) and high (SSP5–8.5, purple) emissions scenarios.

are limited in extent to areas covered by grounded ice. For instance, the >200 m thickness difference in the eastern Kara Sea is caused by differences in ice distribution. CLIMBER2, which drives the legacy run and employs the SICOPOLIS polythermal ice model, simulates a small Eurasian Ice Sheet (EIS) with little ice east of the western Kara Sea at glacial maxima, while in the base run, maximal ice extent crosses the Kara Sea to the Severnaya Zemlya archipelago, inhibiting permafrost formation in that region. The larger EIS footprint in the base run better conforms to observational evidence of EIS extent than the legacy EIS[28], suggesting that adopting the base run ice history may improve the accuracy of the subsea permafrost distribution modeled here. While the GMSL and ice history of the last glacial cycle have the largest impact on present-day subsea permafrost distribution, conditions during the earlier glacial cycles, particularly the penultimate cycle, also affect present-day permafrost thickness and ice content. Overall, using the base ice history decreases the area of seafloor presently underlain by permafrost by 400,000 square kilometers and the mean thickness of that permafrost by 44 m compared to the legacy run.

Though sea level modulates the fraction of time that Arctic sediments spend exposed to air, water, and ice, the variable that drives permafrost formation directly is surface forcing temperature. Mean surface forcing temperature was calculated at each location from the local history of sea-level, ice sheet extent, and air temperature (Fig. 4, see the "Methods" section). Since air temperatures are chosen to be the same in the legacy, base, and GIA runs, changes in surface forcing temperature are driven by varying sea-level curves and ice sheet histories and therefore resemble permafrost thickness changes in Fig. 2B and C.

The change from legacy to base run diminishes temperature forcing—i.e. the mean surface temperatures of the base run are cooler

than those of the legacy run—in much of the Canadian Arctic, the deepest areas of the Laptev and East Siberian Seas, around the New Siberian Islands, and near the White Sea (Fig. 4A). In these regions, subsea permafrost in the base run is thicker than in the legacy run (Fig. 2D). Areas where base run mean temperature forcing is warmer than the legacy run, and subsea permafrost consequently thinner, include the Laptev Sea, islands off the coast of West and East Greenland, and the shallower parts of the East Siberian and Chukchi Seas. Outside of the maximum extent of the Northern Hemisphere ice sheets, differences in forcing temperature between legacy to base are entirely explained by the differences in GMSL. Relative to the legacy run, GMSL in the base run covers shallow continental margin sediments for more time but deep sediments for less time (Fig. 3). Sediments on the shallow continental margin are therefore exposed to higher ocean temperatures for longer, which inhibits permafrost formation, while deep sediments are exposed to seawater temperatures for less time. Within the footprints of the Northern Hemisphere ice sheets, the differences in forcing temperature are explained principally by differences in ice extent. Any place that is covered by ice for longer during the base run than during the legacy run has thinner present-day permafrost because basal ice temperatures are higher than air temperatures; any place with less ice coverage has thicker permafrost.

## GIA effects on present-day subsea permafrost
Present-day subsea permafrost distribution and state are significantly influenced by GIA. The inclusion of GIA in the model reduces the area of the Arctic shelf that is underlain by cryotic sediments at 1850 CE from 2.1 million to 1.8 million square kilometers, i.e. by 14%.

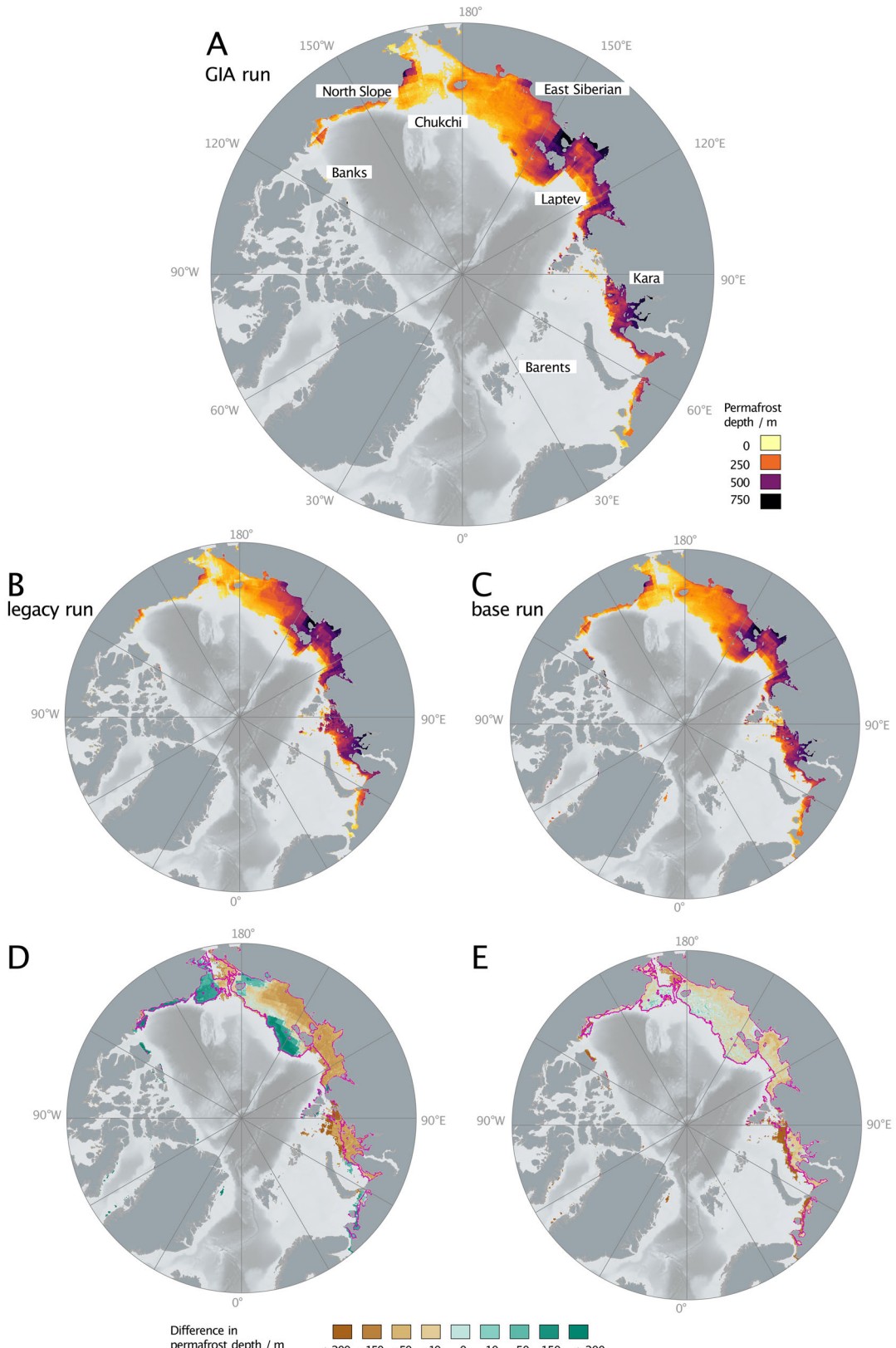

**Fig. 2 | Maps of present-day subsea permafrost thickness. A** Subsea permafrost thickness at 1850 for the glacial isostatic adjustment (GIA) model run. **B** Same as (**A**), but for the legacy model run. **C** Same as (**A**) but with the base run. **D** The difference in permafrost thickness between the base and legacy model runs (i.e. C−B). **E** The difference in permafrost thickness between the GIA and base model runs (i.e. A−C). Areas in **D** and **E** with >200 m difference in permafrost thickness are locations where no permafrost is present in the legacy/base case but permafrost is introduced in the base/GIA cases, respectively. Pink line in **D** and **E** denotes the area of permafrost extent in the GIA run.

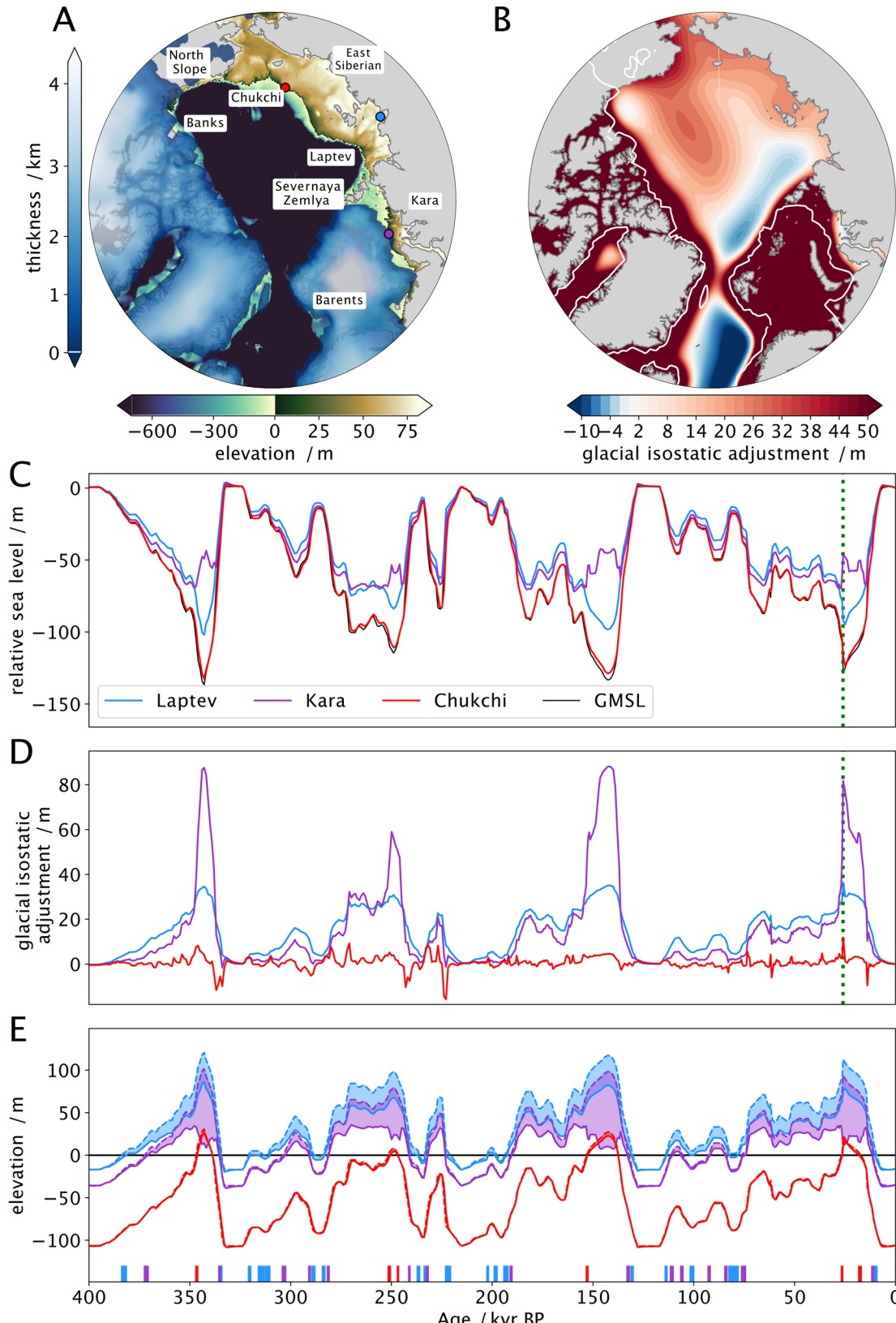

Fig. 3 | Differences between relative and global mean sea levels at example Arctic sites. A Topography of the Arctic continental shelf at Last Glacial Maximum (LGM, 26 kyr BP) in meters above sea level. Colored dots indicate example locations in the East Siberian (red), Laptev (blue), and Kara (purple) seas. Other labeled sites include Banks Island, the Alaskan North Slope, the Chukchi Sea, the Severnaya Zemlya archipelago, and the Barents Sea. Blue colormap indicates the distribution of LGM ice sheets following Peltier et al.[28]. B Difference between relative and global

mean sea level (RSL, GMSL) at Last Glacial Maximum (26 kyr BP). Positive change indicates net RSL rise. C Timeseries of GMSL (black) and RSL at example sites. Dashed green line indicates LGM. D Difference between RSL and GMSL for example sites. E Elevation of example sites. Solid lines indicate elevation including GIA; dashed lines indicate elevation without GIA; the difference is highlighted in solid fill. Vertical dashes indicate times when each site is inundated in the GIA run but not the base run.

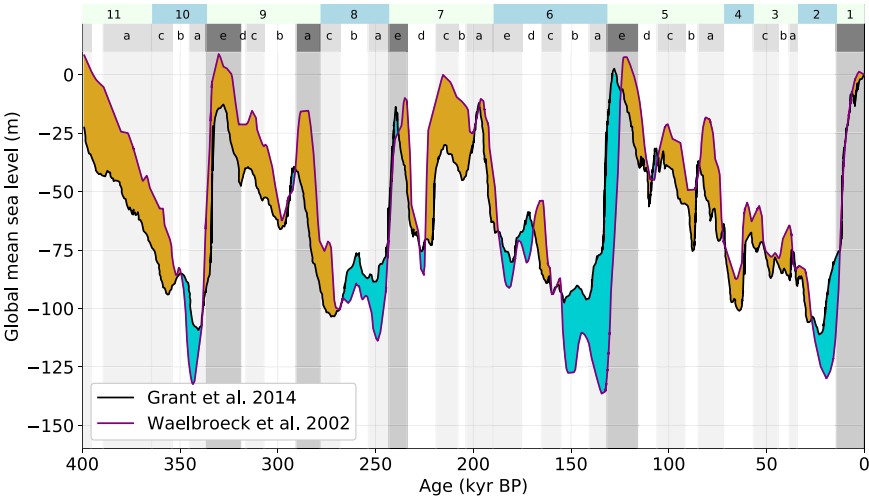

**Fig. 4 | Global mean sea-level curves between 400 kyr BP and present.** Global mean sea-level (GMSL) curves between 400 kyr BP and present are used in the legacy, base, and glacial isostatic adjustment (GIA) runs. Blue filled envelope represents times when the GMSL curve of Waelbroeck et al.[29], used in the GIA and base runs, is deeper than the GMSL curve of Grant et al.[19], used in the legacy run; the brown envelope represents times when the[29] curve is shallower. Numbers and letters along the top edge represent Marine Isotope Stages (MIS) as defined in Railsback et al.[51]. Darker gray bars indicate MIS substages during which substantial subsea permafrost is formed.

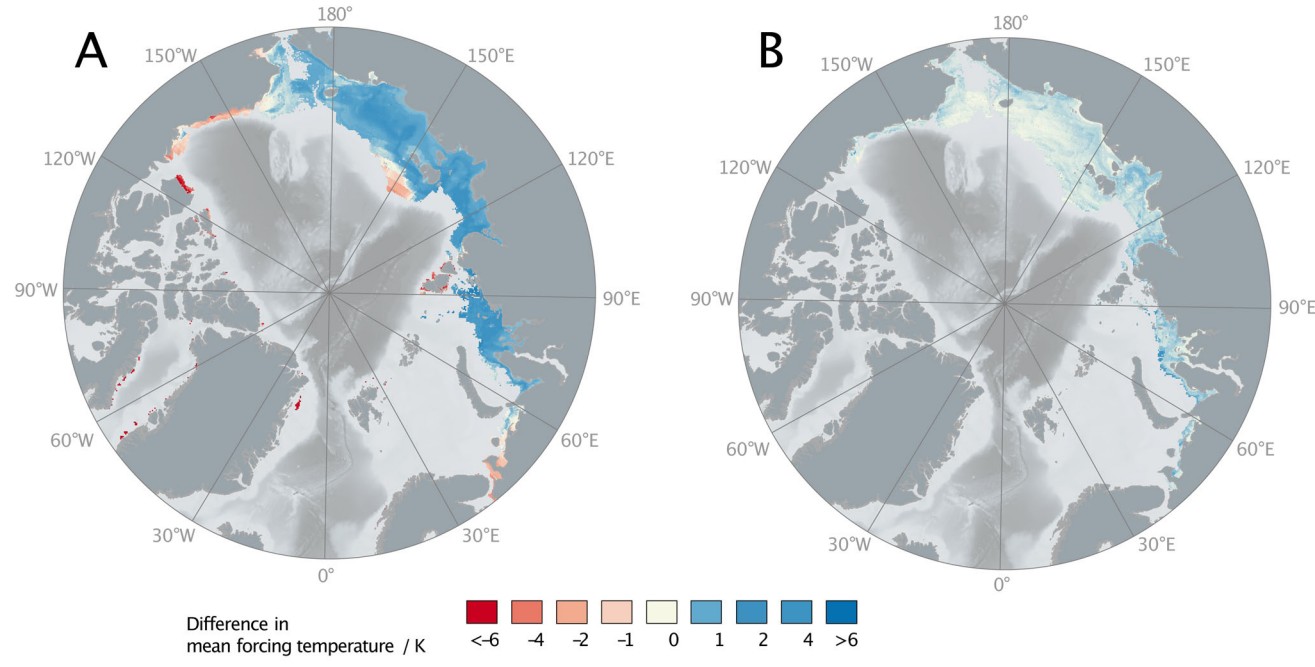

**Fig. 5 | Mean temperature forcing change between subsea permafrost experiments. A** Difference in mean forcing temperature between legacy and base runs. **B** Difference in mean forcing temperature between base and glacial isostatic adjustment (GIA) runs. Mean is taken between 400 ka and 1850 CE. Temperature forcing is the combination of air, ocean bottom, and ice sheet basal temperatures at each grid cell.

GIA causes systematic deviations in RSL on the Arctic continental shelf. These deviations are chiefly due to glacial loading, peripheral bulge dynamics, hydro-isostasy, and gravitational effects. The EIS inhibits permafrost formation in all but the shallowest areas of the Barents and Kara Seas. In those shallow regions where permafrost is present, direct isostatic loading increases sea level when covered by the EIS, as seen in the >80 m rise in GIA in the Kara Sea during glacial maxima (Fig. 5C, D, Supplemental Fig. S1B). Peripheral bulges around the EIS and Laurentide ice lead to negative GIA (RSL is lower than GMSL) and the shape and location of this feature evolves through time (Fig. 5B).

Outside of the peripheral bulge, hydro-isostasy exerts a dominant influence on RSL (Fig. 5B). Hydro-isostasy is the GIA response to changing water load: ice melt during interglacials adds water to the ocean, which depresses the seafloor and elevates continental margins; ice sheet growth during glacials unloads oceans and causes continental margin subsidence[34]. The hydro-isostatic effect is strongest in the Laptev, East Siberian, and Chukchi Seas as well as on the Alaskan North Slope. During glaciations, water unloading leads to the rebound of the oceans and subsidence of continents. Since the water masses rise with the rebounding ocean floor, the sea level at the shelf break follows the global mean while the sea level at the present-day coastline is higher

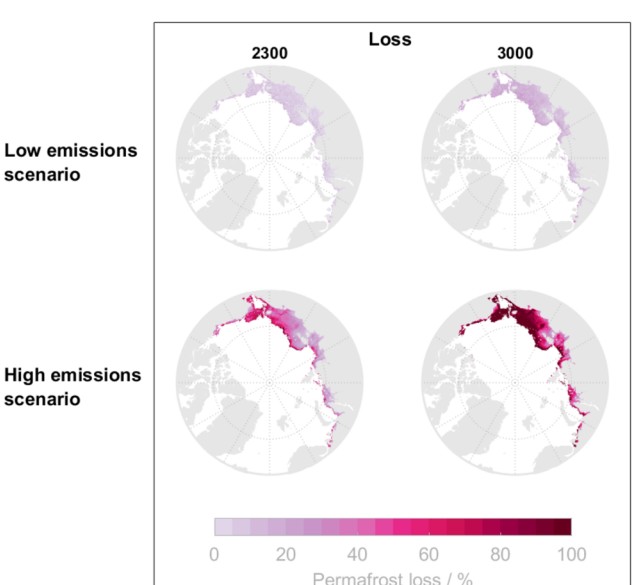
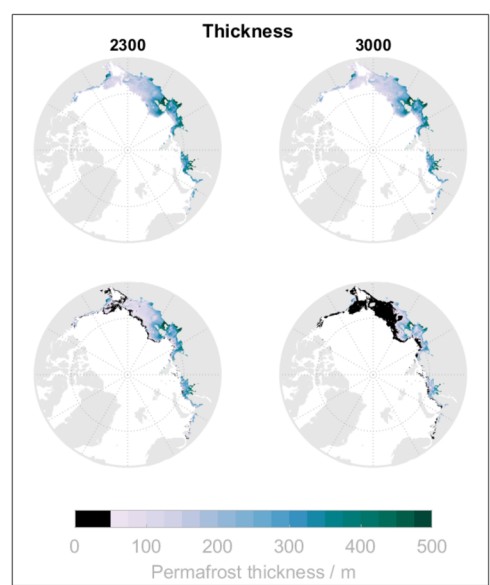

**Fig. 6 | Projected subsea permafrost loss percentage and thickness by 2300 and 3000 CE.** Top row denotes the mean loss percentage and thickness for low (SSP1–2.6) emissions scenarios; the bottom panel is for high (SSP5–8.5) emissions scenarios.

than the global mean (Fig. 5C). This process is reversed during transgressions and interglacials (Supplemental Fig. S1D).

Gravitational effects also have a strong influence on Arctic RSL. Gravity changes RSL in two ways: the change in Earth's gravitational field caused by solid Earth deformation, hereafter deformational gravitation, and the gravitational field generated by the ice sheet itself, hereafter termed self-gravitation[24]. During glacial periods, gravitational effects cause RSL to rise across the Arctic because the self-gravitational effect of the large Laurentide, Eurasian, and Greenland Ice Sheets increases Arctic RSL more than the deformational gravitation effect of ice sheet loading decreases it (Supplemental Fig. S1A). During interglacials, this pattern changes. In areas near the former ice sheets, interglacial gravitational effects cause Arctic RSL to be negative because the self-gravitation of the Northern Hemisphere ice sheets ceases as soon as they melt, while the deformational gravitation effects of the former/smaller ice sheets diminish at the speed of viscous relaxation (Supplemental Fig. S1C). Further from the ice sheets, rotational effects and/or deformational gravitation effects associated with the remnant peripheral bulge cause sea level to be positive in the Laptev, East Siberian, and Chukchi seas.

On average, the GIA run leads to higher sea levels/lower elevations on the continental shelf, which causes mean surface forcing temperatures on the shelf to be higher in the GIA run compared to the base run (Fig. 4B). This causes generally thinner subsea permafrost at present-day in the GIA run compared to the base run (Fig. 2E). Close to the Laurentide and Eurasian ice sheets, the main GIA effect that influences permafrost is direct isostatic loading, which increases inundation (Fig. 5B, Supplemental Fig. S1B). For example, along the western edge of Banks Island and in the Barents and Kara Seas, including GIA causes a thinning of present-day subsea permafrost that ranges from >200 m thinner on the deeper shelf to ~50 m thinner in the shallowest sediments (Fig. 2E). Beyond the peripheral bulge, hydro-isostasy causes cryotic sediment in areas of shallow bathymetries, such as the Laptev Sea, to thin by up to 50 m, while permafrost underlying deeper areas—e.g. distal parts of the East Siberian, Chuchki, and Beaufort Seas—thickens by up to 10 m.

In addition to this broad-stroke GIA signal, temperature and hence permafrost extent also depend on the amount of time that land is exposed. Land exposure time is a function of topography: GIA inundates shallow locations more frequently throughout glacial cycles;

deep locations, only at the beginnings of glacial maxima (Fig. 5E). This leads to the more granular detail in the difference in permafrost thickness between the GIA run and the base run (Fig. 2E).

The amount that GIA reduces subsea permafrost formation depends on solid Earth structure. The GIA-2 run is identical to the GIA run save that the RSL forcing was produced by a GIA model that, relative to the GIA run, has a lower viscosity in the upper mantle but higher viscosity in the lower mantle. This viscosity difference increases overall inundation time, particularly in shallow shelf areas, resulting in >50 m less permafrost in shallow continental shelf areas, 10–50 m less permafrost in deeper areas, and <10 m more permafrost in the deepest shelf areas (Supplemental Fig. S2A). The size of the Northern Hemisphere ice sheets also affects the GIA signal and hence the predicted present-day subsea permafrost. The GIA-3 run is identical to the GIA-2 run save that during MIS-11 to -7 and MIS-5d to -1 (Fig. 3) the GIA-3 run uses ANU ice geometries, which include a larger Eurasian ice sheet than ICE-6G does. This difference increases the Eurasian Ice Sheet's gravitational and deformational influence on the Arctic sea level during these times, leading to 10–50 m less permafrost across much of the Arctic continental shelf (difference between Supplemental Fig. S2A and B).

In total, the inclusion of GIA via the GIA run decreases the area of continental shelf underlain by subsea permafrost by 300,000 square kilometers and the mean thickness of that permafrost by 11 m relative to the base run. The resulting GIA run, seen in the context of GIA-2 and GIA-3 sensitivity tests, is likely an upper bound on the extent and mean thickness of subsea permafrost.

## Future permafrost evolution

The future evolution of subsea permafrost depends on the amount of anthropogenic emissions in the next centuries. Under a low emissions scenario (SSP1–2.6), subsea permafrost as modeled in the GIA run will continue its historical rate of thinning to thin on average by ~30 m to a mean of ~211 m by 3000 CE. This thinning will be concentrated in the central Laptev and Kara Seas due to the thicker present-day permafrost stocks in those areas. With low 21st-century emissions, virtually no areas of seafloor presently underlain by permafrost will completely lose it in the next thousand years (Fig. 6). Under the high emissions scenario (SSP5–8.5), on the other hand, subsea permafrost will thin more than ~38 m everywhere by 2300 CE. This thinning will result in

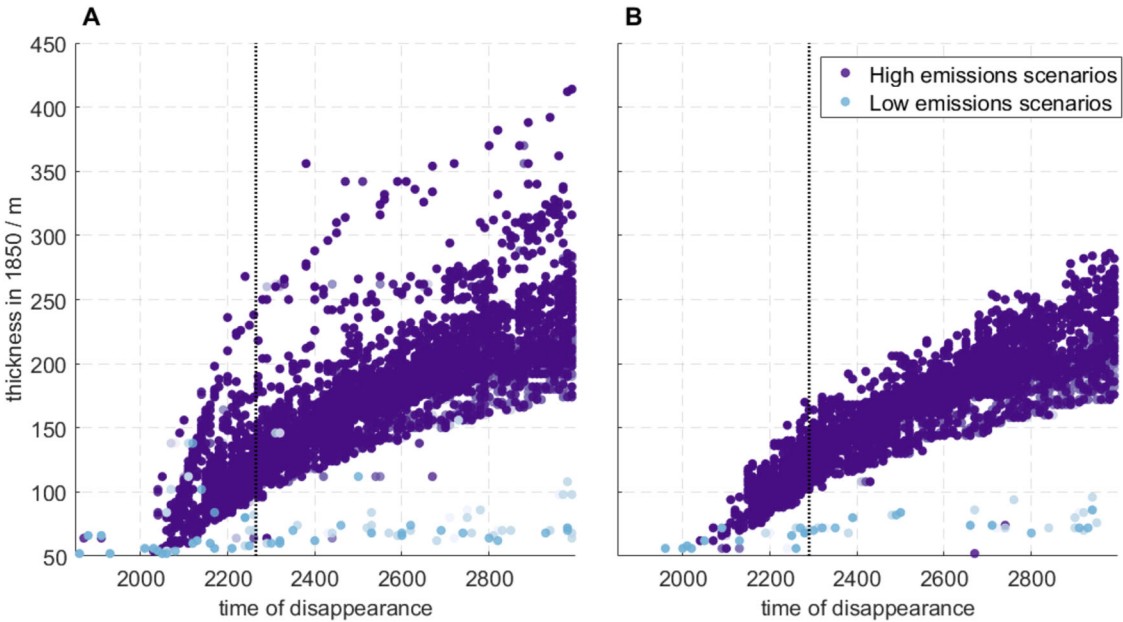

**Fig. 7 | Times of subsea permafrost disappearance.** Time when the permafrost at each location is thinner than 50 m for **A** base and **B** glacial isostatic adjustment (GIA) run. Dashed lines represent the time when all permafrost thinner than 100 m has disappeared.

the disappearance of permafrost—with disappearance defined as permafrost thinning to <50 m—at the outer edge of the Russian Arctic continental shelf and southern Alaska. By 3000 CE, subsea permafrost will have thinned an average of ~153 m, a >60% loss relative to 1850 CE, which will result in subsea permafrost disappearing from the Chukchi Sea, nearly all the Canadian Arctic, much of the East Siberian Sea, and deep areas of the Laptev and Kara Seas.

There is a strong correlation between the pre-industrial thickness of subsea permafrost and its time of disappearance (Fig. 7). Under low emissions, no permafrost thicker than 100 m at 1850 CE thaws before 3000 CE. Under high emissions, all permafrost thinner than 100 m at 1850 CE, but none thicker than 200 m, disappears before 2300 CE. And by 3000 CE, under high emissions, only permafrost more than 160 m thick at 1850 CE remains.

### GIA effects on future subsea permafrost

GIA affects future subsea permafrost in two ways: (1) GIA influence during the late Pleistocene and Holocene leads to thinner present-day subsea permafrost in shallow-water regions (see Fig. 2), thereby reducing the thickness of the permafrost remaining and (2) GIA affects future sea-level change and causes local sea level to differ from GMSL. The former is the significantly more important factor and has been described above. We will expand here on the latter.

Future GIA acts to decrease RSL everywhere on the Arctic shelf, which has a small negative effect on the amount of future subsea permafrost (Supplemental Fig. S3). Less RSL rise decreases the area of newly flooded land, which leads to mean subsea permafrost thickness in the high emissions scenario thinning by ~3 m more by 3000 in the GIA run than in the base run. The GIA effect is modest relative to the GMSL rise, however, which in the projections of Chambers et al.[32] and Greve et al.[35] increases by $8.6\,m \pm 4.6\,m$ by 3000 CE in the high emissions scenarios. During previous interglacials, rising sea levels temporarily increased mean subsea permafrost thickness by increasing the area of inundation. However, when ocean bottom temperatures exceed 0 °C—projected to occur around ~2080 CE with high future emissions[14]—any newly flooded permafrost will rapidly thaw from above as well as below. Beyond this ocean temperature tipping point, future sea-level rise produces no gain in subsea permafrost.

The total effect of GIA causes earlier subsea permafrost disappearance. For instance, all permafrost thinner than 100 m at 1850 disappears ~30 years faster in the GIA run compared to the base run (2260 vs. 2290 CE, Fig. 7). And unlike in the base run, in the GIA run no permafrost thicker than 200 m at 1850 CE disappears prior to 2400 CE (Fig. 7A).

### Discussion

The large influence that different ice sheet histories have on our modeled present-day subsea permafrost distributions highlights the role that late Quaternary ice sheets play in permafrost formation. Ice sheets control permafrost directly beneath them because ice thickness and subglacial hydrology modulate sub-ice temperatures. It has also long been known that terrestrial permafrost can influence ice sheet evolution (e.g.[36–38]). We demonstrate that ice sheets also influence subsea permafrost hundreds to thousands of kilometers beyond their margins because of the gravitational and deformational effects of GIA. This finding supports a growing body of evidence that climatic teleconnections have shaped permafrost evolution in the 20th century (e.g. Romanovsky et al.[39]) and the geologic past (e.g. Li et al.[40]), and will likely continue to do so in the future[41].

Deep uncertainty, defined as uncertainty stemming from disagreement or ignorance about the processes that drive a system, hampers precise projections of sea level over the next century, as do unknowns related to future political decisions[42,43]. Projecting over the next millennium further expands the pool of uncertainty sources. Large uncertainties also surround ice sheet histories for the past four glacial cycles.

While full quantification of these uncertainties is beyond the scope of this study, first steps towards using subsea permafrost to constrain ice sheet histories are already possible using our results. Using the ICE-6G ice history results in thinner permafrost in the Eastern Laptev sea, a finding that better aligns with evidence from seismic surveys suggesting that ice-bonded permafrost exists only in Eastern Laptev sediments coastward of the 60 m isobath[44]. Use of the ICE-6G ice history also increases the modeled thickness and lower boundary of present-day ice-saturated subsea permafrost on the Beaufort shelf (Fig. 2D). This finding better aligns with seismic and borehole data that find the lowermost ice-saturated permafrost in the Beaufort Sea at an

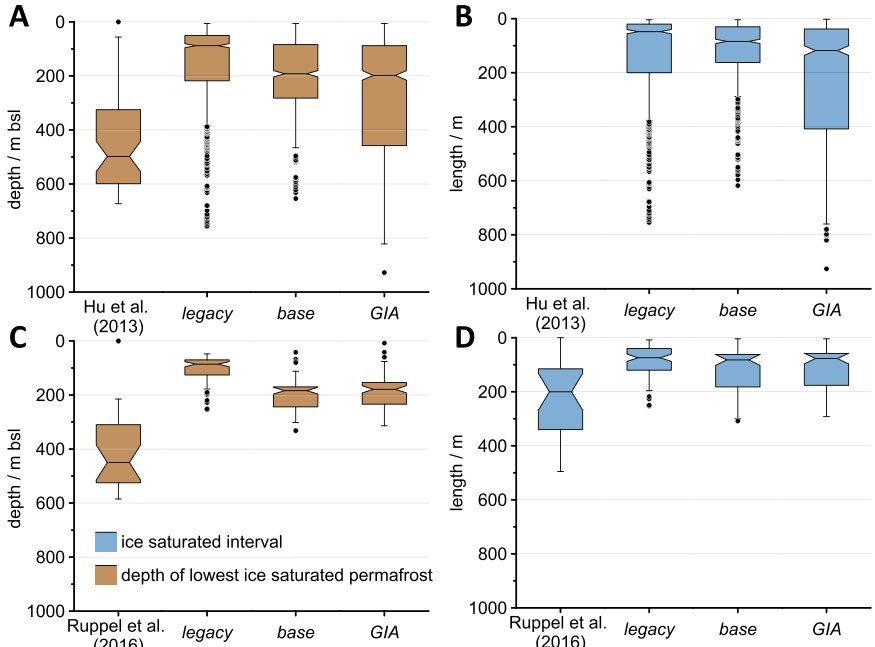

**Fig. 8 | Comparison of modeled permafrost thickness to borehole observations.** Comparison of borehole observations to modeled values for the depth of the lowermost ice-saturated cell (**A, C**) and the length of the depth interval of ice-saturated sediment (**B, D**). Borehole data from the Canadian (**A, B**)[45] and Alaskan (**C, D**)[46] Beaufort shelf regions are compared to modeled values from the three runs (legacy, base, and glacial isostatic adjustment (GIA)) for all modeled locations bounded by the borehole coordinates. Note that the depth interval of ice-saturated sediment is not calculable from Hu et al.[45]. Black dots represent values outside of the interquartile range.

average depth of 500 m (Canadian) and 460 m (Alaskan), and mean thickness of Alaskan Beaufort Sea ice-saturated sediments of 200 m (Fig. 8)[45,46]. Improved data-model fit indicates that the combination of ICE-6G and the GMSL curve of Waelbroeck et al.[29] may represent the Beaufort Sea's history of ice cover, inundation, and subaerial exposure better than CLIMBER2 and the GMSL curve from Grant et al.[19] do. Additional insight is gained from the GIA-3 sensitivity test, which has a larger Eurasian ice sheet and smaller Laurentide ice sheet than the GIA run. Present-day permafrost in the Eastern Beaufort Sea is >200 m thicker in the GIA-3 run than in the GIA run (Fig. S2B), which matches the borehole data of Ruppel et al.[46] even better than the GIA run—evidence to support the smaller Laurentide ice sheet of Lambeck et al.[47].

However, modeled Beaufort Sea permafrost in the GIA run and GIA-2/GIA-3 sensitivity tests are still significantly thinner and shallower than observations. This mismatch suggests that one or more of the forcings used in this study—including ice sheet geometries, RSL, and air temperature—are imperfectly representing the region's late Quaternary history; or that subsea permafrost in this region may be influenced by processes not accounted for in our model. Processes that our model omits include permafrost formation beneath shallow ice sheet margins, spatial variations in benthic temperatures driven by the inflow of warm Atlantic water into the Arctic, changes in river and drainage basins, and spatiotemporally discrete sedimentation and erosion events such as glaciogenic debris flows, the transgression of which would produce additional syngenetic subsea permafrost. Though the inclusion of these factors exceeds this study's scope, they likely have significant impacts on subsea permafrost formation and should be included in future pan-Arctic permafrost models.

Beyond the Beaufort and Eastern Laptev Seas, the lack of observational constraints leaves the updates in subsea permafrost distribution made here open to future observational ground-truthing. Such is the case off the west coast of Banks Island, Canada, where our GIA run predicts no subsea permafrost but Overduin et al.[18] map subsea permafrost that in places exceeds 200 m. Should future observational campaigns target regions such as Banks Island or the eastern Kara Sea, they will have the added benefit of constraining not only subsea permafrost itself but also the local glaciation histories of the Eurasian and Laurentide ice sheets.

Future work should explore subsea permafrost's sensitivity to lateral variations in lithospheric thickness and mantle viscosity. The sensitivity tests that we conducted (GIA-2 and GIA-3 runs) demonstrate that a 1D GIA model with stronger lower mantle viscosity but weaker upper mantle viscosity than the structure used in the GIA run produces RSL fields that inundate the shallow Arctic continental shelf for a greater fraction of the late Quaternary. There is some support for stronger lower mantle viscosity in the Arctic: RSL predicted using a 1D Earth structure with a stronger lower-mantle viscosity (VM7) was found to improve GIA model fit to postglacial Arctic RSL data[48]. There is also evidence that 3D viscoelastic Earth structure has had significant impacts on Arctic RSL over the last two glacial cycles. Laterally varying solid Earth structure was found to increase postglacial RSL in the White and Kara Seas, but have a variable effect in the Laptev Sea, with higher RSL during the Holocene but lower RSL earlier in the postglacial period[48]. Another study[49], which focused on the Last Interglacial (LIG, 130–116 kyr BP), found that lateral viscosity variations increased RSL throughout the LIG for parts of the East Siberian, Chuckchi, and Eastern Laptev Seas but decreased RSL in the western Laptev and Kara Seas. However, there are large uncertainties in 3D viscoelastic Earth structure in the Arctic. These uncertainties, combined with the prohibitive computational expense of simulating Arctic sea level over multiple glacial cycles, make it currently impossible to assess the effect of 3D Earth structure on permafrost. Rather, these limitations motivate efforts to estimate Arctic Earth structure, which, when combined with advances in computing, would also enable better estimation of subsea permafrost.

Future work should also focus investigation of the sensitivity of present-day permafrost to ice sheet variations during times when ice histories are especially uncertain. Those times include the LGM, where ice sheet modeling continues to disagree with sea level estimates of

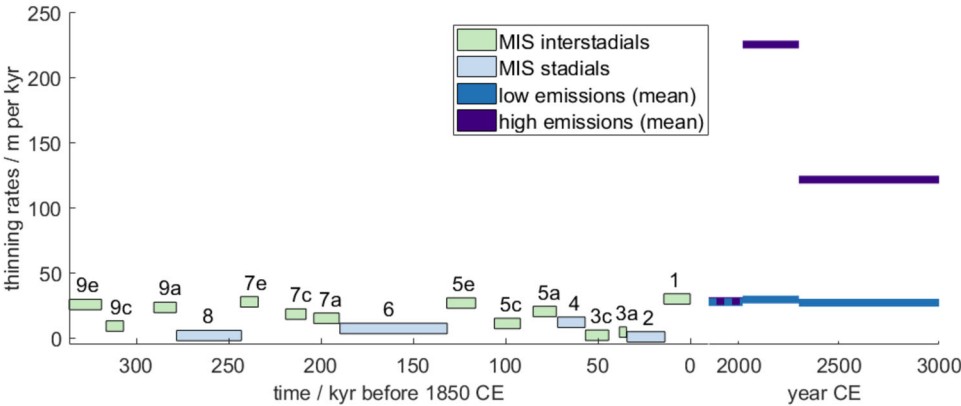

**Fig. 9 | Comparison of past rates of mean subsea permafrost thickness change to future change rates.** Mean subsea permafrost thickness change rates between 400 kyr BP and 3000 CE. Horizontal lines denote mean rates of subsea permafrost thinning for each Marine Isotope Stage (MIS) during which subsea permafrost existed and for future predictions.

global ice volume[50]; MIS-3 [57–34 kyr BP][51], when recent evidence suggests GMSL may have been more than 20 m higher than modeled here[29,52–54]; and the penultimate deglaciation, when the size of the EIS and its collapse history remain largely uncertain[55]. Future work could also test subsea permafrost's sensitivity to the history of the Siberian ice sheet, which during the penultimate and earlier glacial cycles may have held significant mass[56]. Differences in ice sheet loading during these intervals, and the accompanying sea-level variations, would produce characteristic spatial signatures in present-day permafrost. This line of inquiry points to subsea permafrost as an as-yet-untapped constraint on past ice-sheet histories.

The analysis of subsea permafrost presented here has implications for the amount of organic carbon that subsea permafrost presently holds and therefore its potential as a future emitter of greenhouse gases. Structured expert assessment of subsea permafrost places present-day stocks of organic carbon and methane, respectively, at ~560 (170–740, 90% confidence interval) and 45 (10–110) gigatons of carbon, and projects that subsea permafrost could emit 190 (45–590) gigatons $CO_2$-equivalent[11]. Recent work has revised that estimate to a higher value of 2822 (1075–5963, 90% confidence interval) gigatons of remaining organic carbon stocks stored in subsea permafrost[13]. Our work suggests that present-day subsea permafrost is thinner than previously thought in shallow regions and in the western Russian Arctic, in some areas by several hundred meters. We also find that the area of seafloor presently underlain by subsea permafrost, as estimated in the GIA run, is >25% smaller than previously estimated[18]. These findings reduce both the amount of organic carbon that subsea permafrost may hold and the total amount of greenhouse gases that it may, through future thaw, release, though lack of consensus remains about what proportion of the $CO_2$ and methane released by subsea permafrost reaches the atmosphere[57,58]. The rate at which subsea permafrost may release methane is also largely unknown. If ocean bottom warming can more easily destabilize methane associated with thinner subsea permafrost, our findings may increase the near-term climate risks that subsea permafrost thaw poses. Projecting into the future, our results constrain the spatial distribution of future permafrost loss as well as the pace of its thaw. These findings can inform present planning for future community-based and industrial undertakings on the Arctic continental shelf, as such activities rely on accurate assessment of subsurface sediment characteristics.

Comparison of future climate projections with paleoclimatic analogs can give perspective on the effect that human activity has had on the climate system. We provide this context by comparing our projected rates of future subsea permafrost thinning to thinning rates over the past four glacial cycles (Fig. 9). Mean rates of past subsea

permafrost thinning during interstadials have ranged from 5 m per kyr during MIS-9c to 31.2 m per kyr during MIS-7e (Fig. 9). In previous interglacial periods during which average subsea permafrost thickness exceeded 200 m, e.g. MIS 9e, 9a, 7e, 5e, 1, subsea permafrost thinned at an average rate of ~27 m per kyr. We project that subsea permafrost will thaw at a rate similar to 1850 speeds until 2050 (29 m per kyr) regardless of the emissions scenario. In the second half of the 21st century, human activity will have a significant effect on subsea permafrost thinning rates. Under low emissions scenarios, the present-day rate of thinning continues to 3000 CE. High 21st-century emissions, however, will accelerate thinning between 2050 and 2350 CE to >8 times faster than the fastest thinning rate since MIS-9. Between 2350 and 3000 CE, thinning rates remained at 110 m per kyr, which is roughly four times faster than pre-industrial values.

Subsea permafrost thaw accelerates under the high emissions scenarios because the Arctic passes a climate tipping point. Loss of Arctic sea ice, included in our model via the modeled bottom water temperatures from Wilkenskjeld et al.[14], spurs the Arctic to warm at a rate faster than the global mean[59]. The positive feedback loop inherent in Arctic amplification—wherein lost sea ice lowers Arctic albedo, which increases sea ice loss—leads to cascading effects on the Arctic climate system. These effects include the warming of Arctic shelf waters above zero degrees[14], a tipping point past which subsea permafrost thaw accelerates from both above and below. Though this acceleration is avoided under the low emissions scenario, under the high emissions scenario the tipping point occurs at ~2080 CE.

Our new pan-Arctic simulation of subsea permafrost from 400 kyr BP to 3000 CE enables an updated assessment of the history, present-day characteristics, and future evolution of subsea permafrost that accounts for the effects of GIA. We find that GIA influences subsea permafrost evolution everywhere on the continental shelf, with the deformational effects of ice sheet loading dominant in the Barents, Kara, and Beaufort Seas, and hydro-isostasy dominant in the Laptev, East Siberian, and Chukchi Seas. Our new subsea permafrost map, based on the GIA run, has 14% less seafloor area underlain by permafrost and is 4% thinner than the base run. Both the GIA and base runs update the ice cover and sea level forcing of the legacy run (cf. Overduin et al.[18]), resulting in even less permafrost: the base run has 14% less area and is 8% thinner than the legacy run. Sensitivity tests (GIA-2 and GIA-3 runs) suggest that the GIA run may represent an upper bound on subsea permafrost extent and thickness.

The recent IPCC-AR6 suggests that future permafrost thaw would be insufficient to trigger self-reinforcing acceleration in climate warming[60]. The same is not true of the future effects of climate warming on subsea permafrost. Under a high emissions scenario that

includes the loss of year-round Arctic sea ice, which is included in our modeling, self-reinforcing feedback in the climate system triggers a rapid, irreversible acceleration of subsea permafrost thaw that begins in the next 60 years and persists so long as ocean bottom temperatures exceed 0 °C. This possible future adds yet more urgency to efforts to slow human emission of greenhouse gases in the next quarter century.

## Methods

### Permafrost model

Permafrost extent and composition were calculated from the output of a 1-D heat transfer model. We used CryoGrid 2, a 1-D heat diffusion model introduced by Westermann et al.[61], which is a model that continues to develop. The current version is described in a release paper[62] and the code is available at https://github.com/CryoGrid/CryoGridCommunity_source/releases/tag/GMD (accessed 20.05.2022). The model was implemented similarly to the implementation in Overduin et al.[18], save that we changed the synthesized forcing temperature by using different sources for sea level, ice sheet histories, and began the model at 400 kyr BP rather than 450 kyr BP. We performed calculations at grid cell centers of the 12.5 km EASE-Grid 2.0[63] and included any locations with present-day elevations between 187 m below and 18 m above sea level (bsl, asl)[64]. Sea level—either RSL (GIA runs) or GMSL (legacy and base runs)—was combined with the IBCAO 4.0[64] bathymetric map to produce paleotopography, which was used to determine the water depth or exposure of each grid cell.

The lower boundary condition for permafrost was temporally invariant heat flux drawn from the globally distributed data of Davies[65]. The upper boundary condition was temperature, either land surface, seabed, or subglacial, as described in the following.

Historical land surface temperature was forced with air temperature from the CLIMBER2 intermediate complexity Earth System Model[20]. Under conditions of future sea-level change, some modeled locations may submerge or emerge, and thus require forcing with future land surface temperatures until submergence or following emergence. This applied to only a few locations in our modeling domain, usually next to the coast. In these few cases, constant temperatures equivalent to those during pre-industrial times (1850 CE) were applied. Though permafrost was removed from present-day locations where warm bottom water in deltaic and estuarine settings likely precludes permafrost formation, no assumptions were made about the locations of paleo rivers and estuaries. This likely results in a minor overestimation of subsea permafrost in those regions.

Historical seabed temperatures were forced as a function of water depth, based on observational data from the Siberian shelf area[66]. Reductions in sea ice cover extent and duration are expected to warm the seabed since brine produced by freezing sea ice cools the seabed. Wilkenskjeld et al.[14] shows warming of the seabed by up to 10 °C under more severe climate change scenarios such as SSP5–8.5 (Supplemental Fig. S4). The increase in seabed temperatures is strongly related to the disappearance of sea ice. Our future seabed temperature forcing was adjusted by the spatial-mean anomaly of projected decadal mean seabed temperatures for either a low (SSP1–2.6) or high (SSP5–8.5) emissions scenario[14] from 1850 to 2950 CE, consistent for each run with the corresponding ice sheet model forcing (Table S1). Temperatures from 2950 to 3000 CE were held constant at the 2950 CE level. Subglacial temperatures were treated as warm-based for ice masses exceeding 100 m in thickness and set to 0 °C, as in Overduin et al.[18].

### Glacial isostatic adjustment model

GIA was calculated following the algorithm of Kendall et al.[67], which computes gravitationally self-consistent sea-level variations that are caused by ice and liquid water loading on a viscoelastic earth.

Calculations include the effects of shoreline migration and the impact of load-induced Earth rotation changes on sea level[68,69]. The resulting calculation yields a spatiotemporally continuous estimation of RSL that is linearly interpolated to the grid centers of the EASE-Grid 2.0 and combined with the IBCAO 4.0 bathymetry. For the GIA run, we assume a radially symmetric viscoelastic Earth structure with a viscosity following the VM5 profile[28] and the elastic structure and density from the PREM seismic model[70]. For the GIA-2 and GIA-3 runs, we adopt an Earth structure with a lithospheric thickness of 71 km, upper mantle viscosity of $3 \times 10^{20}$ Pa s, and lower mantle viscosity of $50 \times 10^{21}$ Pa s.

### Ice histories

Our ice history from the Last Glacial Maximum (LGM) to 1950 CE for the GIA run follows ICE-6G[28]. The ICE-6G history was then extended back over four glacial cycles following the GMSL curve from Waelbroeck et al.[29], which is based on RSL observations and $\delta^{18}O$ records from benthic foraminifera (Fig. 3). Ice sheet geometries prior to the LGM were chosen by finding the post-LGM ICE-6G geometry that best matches each pre-LGM GMSL value. For GMSL values prior to LGM that fall outside of the range of LGM to present values, we assume the closest available GMSL value. This assumption resulted in a present-day GMSL during MIS-9e and 5e since no template of pre-LGM ice collapse is available in the ICE-6G deglacial history. Though GMSL during these times was higher than present-day GMSL (e.g. de Gelder et al.[71]), this approximation is expected to have a negligible effect on the results presented here. There is evidence that the ice sheet configuration during the penultimate glacial maximum differed significantly from that during the last glacial maximum[72]. We therefore followed the approach of Dendy et al.[55], replacing the EIS geometries between 200 and 130 kyr BP with reconstructions from Lambeck et al.[25,31] and pairing them with Laurentide ice sheet geometries chosen from the post-LGM ICE-6G history in order to maintain the GMSL curve of Waelbroeck et al.[29]. For the GIA-3 run, the ANU Eurasian[25], Laurentide[47], and Greenland[73] ice sheet histories from LGM to present were combined at each timestep with an Antarctic ice geometry from the ICE-6G ice history in order that the overall ice volume of this history, hereafter ANU-6G, match ICE-6G's LGM-to-present volume. The ANU-6G history was then extended back over four glacial cycles following the procedure outlined above.

For the ice geometry between 1950 and 2015, we used the ice thickness from the Ice Sheet Model Intercomparison Project (ISMIP6)[74,75]. ICE-6G's 1950 CE ice extent is not in full agreement with the 1950 CE ice thicknesses from ISMIP6[74,75]. We therefore constructed a smooth transition from ICE-6G to ISMIP6 ice extents by tapering the difference between the two models from 0% to 100% between 0 CE and 1950 CE, then added it to ICE-6G. The GIA simulation was run from 400 kyr BP to 1950 CE with timesteps of 100 yr, which were interpolated using nearest neighbor interpolation to the 100 yr timesteps of the permafrost simulation and linear interpolation to the 12.5 km spatial resolution of the EASE-Grid 2.0. The ANU-6G history was not extended past 1950 CE.

Between 2015 and 3000 we used an ensemble of 17 Antarctic and 14 Greenland ice models from the SICOPOLIS polythermal ice-sheet model[35,76], which, following the ISMIP6 protocol, were produced with dynamic oceanic and atmospheric forcing between 2015 and the end of 2100 and constant forcing through 3000. See Chambers, Greve, et al.[32,33] for full details on Antarctic and Greenland, respectively. AIS and GIS ensemble members with identical generalized circulation model (GCM) forcing, ocean forcing, and emissions scenario (SSP/RCP) were paired. AIS members with no identical GIS analogs were paired with a GIS member produced by the same emissions scenario. See Table S1 for details on the list of GIS/AIS pairings. The GIA simulation was run with timesteps between 10 and 100 yr, which were interpolated using nearest neighbor interpolation to the 10 yr timesteps of the permafrost simulation.

## Permafrost model output and data analysis

Model output included subsea permafrost thickness and ice content at 2 m vertical spacing over depth to 2 km below the land surface or seabed, at the modeled EASE Grid 2.0 locations. The temporal resolution of the output is 100 yr for the historic period until 1850 CE and 10 yr for the future projections.

The model was run over all possible permafrost locations, i.e. all locations on the EASE Grid 2.0 with present-day elevation between −187 and 18 m asl as this encompasses the maximum range of RSL change in the forcing data. We also applied a filter to rule out locations in big river deltas and estuaries, including grid cells near the Ob and Lena rivers, St. Petersburg Gulf, the Baltic Sea, near Iceland, south of Kamchatka, and the Bering Strait. This filter is applied because the permafrost in those locations is likely mis-estimated due to its dependence on estuarine sedimentary processes which are not represented in our model. Results were then further filtered, to include only locations that (a) have been subaerial for at least 100 yr during the model period, (b) are currently submerged, and (c) have present-day permafrost deeper than what a theoretical present-day steady-state solution yields (cf. Overduin et al.[18]).

To evaluate possible future thinning rates of subsea permafrost, we calculated the mean projected thinning rates within the low (SPP1–2.6) and high (SSP5–8.5) emissions scenarios for the historic period (1850–2020 CE), the near future (2020–2300 CE) and the distant future (2300–3000 CE). For comparison, we calculated the mean thinning rate between minimum and maximum mean permafrost thickness for each MIS.

We compare our modeled lower permafrost bound to observations determined using a combination of well-log and temperature records from the Beaufort shelf (Canadian[45]; Alaskan[8]). Most well-log records vary as a function of ice saturation of the sediment pore space (e.g. bulk sediment propagation velocity or electrical resistivity), whereas our modeled values reflect the depth of the 0 °C isotherm. Values from Ruppel et al.[8] are based on their assessment of intermediate ice saturation; only permafrost lower limit observations of high data quality (i.e. the a or b categories from Hu et al.[45]) were included. All modeled grid cells within the longitudinal range covered by the industry wells are included, i.e. from the coastline out to the outer edge of permafrost occurrence. The proximity of the industry wells to the shoreline skews to thicker permafrost.

Observed lower bounds of permafrost are deeper than the differences our models predict, and differences between the model runs are smaller (<55 m) than between the mean modeled and mean observed (298 m, Ruppel and 274 m, Hu).

## Data availability

All data needed to reproduce figures, including input GIA and ice models, are available on Zenodo (https://doi.org/10.5281/zenodo.10499329).

## Code availability

The GIA modeling is available at https://github.com/jaustermann/SLcode. Code used to produce the permafost modeling is adapted from Cryogrid 2 (https://github.com/CryoGrid/CryoGridCommunity_source/releases/tag/GMD).

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

## Acknowledgements

This work was supported by the National Science Foundation grant OCE 18-41888 (R.C.C., J.A.). The project has received funding under the European Union's Horizon 2020 Research and Innovation Program under grant agreement no. 773421 (F.M., P.P.O.). We (J.A., R.C.C.) acknowledge computing resources from Columbia University's Shared Research Computing Facility project, which is supported by NIH Grant 1G20RR030893-01 and NYSTAR Contract C090171. We (F.M., P.P.O.) acknowledge support by the Open Access Publication Funds of Alfred-Wegener-Institut Helmholtz- Zentrum für Polar- und Meeresforschung.

## Author contributions

Conceptualization: R.C.C., J.A., F.M., P.P.O.; Formal analysis: R.C.C., J.A., F.M., P.P.O.; Funding acquisition: J.A., P.P.O.; Investigation: R.C.C., J.A., F.M., P.P.O., S.W.; Methodology: R.C.C., J.A., F.M., P.P.O.; Software: R.C.C., J.A., F.M., P.P.O.; Visualization: R.C.C., F.M., P.P.O.; Writing—original draft: R.C.C., F.M., P.P.O.; Writing—editing: R.C.C., J.A., F.M., P.P.O., S.W.

## Funding

## Competing interests

The authors declare no competing interests.
