## [Peer Review File · Nature Communications]

Glacial Isostatic Adjustment reduces past and future Arctic subsea permafrostEditorial Note: Parts of this Peer Review File have been redacted as indicated to remove third-party material where no permission to publish could be obtained.

Reviewer #1 (Remarks to the Author):

Review of paper NCOMMS-23-09654: " Glacial isostatic adjustment reduces past and future arctic subsea permafrost"

I would like to say to the authors that I very much enjoyed reading this paper. It is well written. I suggest it is accepted for publication with some minor changes I list below. They are mostly related to minor changes to terminology and additional information about the method section.

Line 133 "Permafrost that exceeds a thickness of 500m also underlies the shallow central Kara Sea.." Should this not be subsea permafrost, as in the rest of the paragraph.

Section 2.1.1

When comparing the legacy and base model runs - two factors have been changed: ice sheet history (CLIMBER vs ICE5G) and GMSL curve (Grant and Waelbrook).

To entirely isolate the impact of just GMSL, did the authors consider using the CLIMBER model and Waelbrook curve. As the set-up stands, I am not sure how to separate the ice sheet history and GMSL. Line 157 " Overall using the base ice history decreases the area of the seafloor presently underlain by permafrost..... " Is this the choice of ice history or the GMSL curve?

Line 167 "The change from the legacy to base run diminishes the temperature forcing...." Why does this decrease? Can you expand on why the temperature is cooler in the base run? I have read through the Methods and the description of the temperature calculation is very brief. For example line 420: " Reduction in ice cover extent and duration are expected to warm the sea bed". Therefore, if the base run produces a cooler temperature is this due to a reduction in ice cover, increased water depth?

Choice of Greenland ice sheet: Both the CLIMBER and ICE6G Greenland ice sheet reconstructions are 'small' (as on Fig.5A) and do not really extend beyond the present-day margin. Would a large Greenland ice sheet impact on your results? - as the continental shelf around Greenland would have a fluctuating ice sheet extent?

Section 2.1.2:

Line 181 " Gravitational effects tend to be smaller since the rebounding earth in part counteracts the gravitational effects..."

I do not quite follow this sentence. Do you mean the gravitational effects are smaller than the deformational effects, driven by the rebound (shown in Fig. S1b)

Line 189 "Outboard of the peripheral bulge... which depresses the oceanic crust..." This is an odd terminology, do you mean outside? Beyond? I think it would be more general to say the sea floor, or ocean bed, as it is not just the crust which is depressed.

Line 202: "Inboard of the Laurentide and Eurasian peripheral bulge". . Again Inboard is odd terminology, Inside, Close to?

Line 207 : Reference to Fig. 2A - I think should be Fig.2D and E.

Section 2.2:

Great to see the future projections continued to 2300 and 3000 yr.

Fig. 6. Nice investigation of the relation between 1850 thickness and time of disappearance. However, the dark blue dots which make up most of the figure - I am not sure if these relate to high emission (pale purple) or low emission (pale blue)

Line 242: "GIA affects future... (1) 400,000 years of GIA influence leads to thinner present-day subsea permafrost" How are you assessing that this is 400,000 years of GIA and not 2 glacial

cycles or 1 glacial cycles?

Line 245: "causes local sea level to differ from the mean" revise to "differ from GMSL, to tie in with the rest of the paper terminology.

Section 3:

Line 270: "We demonstrate that the ice sheets also influence.... kilometres distant due to the gravitational"

Please change this wording - kilometres distant due to, does not make sense.

Line 282: "harnessing subsea permafrost as an ice sheet constraint" harnessing is an odd terminology, please revise .

Fig. 8 - nice figure, but can you label what the black dots are?

Line 296: "However, modelled Beaufort Sea permafrost in the GIA run is still significantly thinner ... may be influenced by processes not accounted for in our model". Rather than this misfit been related to a missing process - why could it not be due to problems with the choice of ice sheet reconstruction. CLIMBER model is not constrained to observational data and ICE6G is just one possible reconstruction. On Fig. 2 the extent of permafrost in the Beaufort Sea is very small or minimal (in the region of 30W to 90W). Are the authors referring to 120W- 170W?

Fig 9: nice figure.

Line 366: "hastens the sea ice loss" hastens is not a correct word. I would use increases the sea loss or some other phrase.

SOM. Fig S3 - please reference in the main paper.

Methods section:

The description of each model, section 5.1 and 5.2 was fine. However, can you expand on how the two models are linked? On line 398 "changed the synthesized forcing temperature by using different sources for sea level, ice sheet histories a..." So, what variable/output from the GIA model is included in the permafrost model in the GIA-runs. Do you use a prediction of RSL? Paleotopography?

Line 162 " Mean Surface forcing temperature were calculated at each location from the local history of sea-level, glacial load and air temperature "

Can you clarify this in the method section. Depending on the choice of model (base, legacy or GIA)I assume the 'local history of sea level' is either GMSL of Grant, Waelbrook or RSL from GIA model. What is glacial load? Do you mean ice sheet extent or the deformation signal from the GIA model? How is this incorporated in the "non" GIA run.

I think this would be clearer if in the Method section you defined clearly what was the input to the permafrost model and where it was from sourced from, perhaps in a table,

How do you regrid the GIA model output to the higher resolution of the permafrost model (line 400 12.5km)? On line 466 "interpolated using nearest neighbour interpolation to the 100a timestep of the permafrost simulation" , the authors define how the account for the differences in temporal resolution but not the spatial resolution.

Line 482" Model output included sediment temperature and composition'.... A main output that is

evaluated in the paper is thickness, so I would list this as a main output. Sediment temperature and composition are not evaluated, or discussed in the paper.

“defined modern elevation”

Line 110 “permafrost was modelled between 136m below and 18m above present day sea level”

Line 402 “included any locations with modern elevations between 187m and 18m above sea level”

Line 487 “all locations on the EASE Grid.. with current elevations between -138m and 18m asl”

Can you check these numbers and definitions to be consistent? For example, ‘modern’ or ‘current’. I am assuming the 136m is the GMSL maximum, but the RSL variations will be larger? What was the reason for + 18m,+187m?

Figures:

Fig.1: “GMSL from Waelbroeck et al., 2002 and Peliter et al., 2015”. I am assuming that the GMSL curve plotted is from the updated ice history described in Section 5.2, as Peltier et al., 2015 does include 400ka results. From reading the caption, I was looking for multiple lines.

Fig. 2: Can you add ‘A-GIA run’, ‘B -legacy run’ and ‘C- base run’ to the figure itself. To make the difference plot clearer and to link with the caption “> 200m difference in the permafrost thickness are locations where additional permafrost is introduced” could you add a contour to mark of the edge of the permafrost extent on D and E from the legacy run. It would make it easier to identify where there are new areas of permafrost are simulated in the base and GIA runs.

Fig.4: Can you reverse the colour scale. The colder temperature red and warmer temperatures red is counter intuitive.

Fig 5a: It is not possible to identify the “ice regions”, with the blue colour scale, which looks the same as the negative elevation regions. Can you change the ice covered areas to be different colour shade, and add a scale bar.

Fig 5b and S1: With the colour scale chosen it is not possible to identify the differences between - 8m or -4m. Please modify. Can you choose a more contrasting colour? Can you add a contour line for Fig. 5b to mark the edge of the ice sheet regions? As the text describe the regions of peripheral bulge at the edge of the EIS and Laurentide, this will help the reader to identify the edge of the ice sheet. Change exemplary (which refers to something been excellent) to example, or reference.

Can you add a simple table in the SOM of “Model reference name (Legacy, base), ice sheet history, GMSL curve, with/without GIA”.

Reviewer #2 (Remarks to the Author):

Review of the manuscript “Glacial Isostatic Adjustment reduces past and future Arctic subsea permafrost” by Creel et al.

The manuscript incorporates the influences of glacial isostatic adjustment (GIA) in the Pan-Arctic model of subsea permafrost for the last 400,000 years and extends the simulation 1000 years into the future for different SSP scenarios. They find that the incorporation of GIA can reduce the present-day subsea permafrost thickness and the subsea permafrost is preserved under a low emissions scenario but mostly disappears under a high emissions scenario.

The paper is generally adequately presented and well organized. The figures are of good quality. However, the GIA doesn’t consider the 3D Earth structure, which has been shown having significant impacts on sea-level predictions, especially near the ice-covered regions like Arctic (Austermann et al., 2013; Kuchar et al., 2019), neither the uncertainty related to ice model is considered (e.g., Melini & Spada, 2019). For example, in Fig. 2, the difference between base and legacy model runs is larger than the difference between base and GIA models runs, indicating use of different ice history and GMSL has larger influence (than the incorporation or GIA or not) on the

subsea permafrost thickness results at 1850. Moreover, the ice history uncertainty is dramatic in the Arctic region (e.g., Patton et al., 2015, 2017), which needs to be investigated. The authors state the GIA influence on subsea permafrost distribution and state is significant (14%) in the abstract, line 176 and conclusion, while I wonder if the uncertainty is considered, whether the GIA influence is still statistically significant and by how.

Detailed comments:

1, Line 18, "International Panel on Climate Change's sixth assessment report" should be changed to "Intergovernmental Panel on Climate Change sixth Assessment Report". Same in line 94.

2. Line 101, change "(Waelbroeck et al., 2002)" to "Waelbroeck et al. (2002)"

3. In Fig. 2, label the locations. Although you commented "see Fig. 5A for locations", it is not readers-friendly to go back and forward and the locations are already referred in line 132-135. And in the fig caption, the last sentence "Areas ..." is not clear to me, does it mean the areas with >200 m difference are the places that in legacy run have no subsea permafrost? If yes, better to distinguish with different color, as based on the legend, the colors are mainly to show the depth difference, can use other color/map to indicate the subsea permafrost geographical coverage difference between different model runs.

4. Line 142, the ">500 m" cannot be identified in the plot. Change "fig. 2B" to "Fig. 2D".

5. Line 171, change "2B)" to "(Fig. 2D)". And in line 173, change "include" to "including".

6. Line 178, & 217-218, how about the differences between legacy and base runs? Both for the subsea permafrost coverage area and mean thickness.

7. Fig. 6 caption, "loss percentage and thickness", does the thickness refers to the remaining permafrost thickness? If yes, may add a "remaining" before "thickness".

8. Line 260-261, 2260 vs 2290, shouldn't be the difference ~30 years? Please label 2260 and 2290 in Fig 7 (maybe a vertical line to mark the year).

9. Line 332-333, both "gigatons" and "Gt" are used, needs to be unified.

10. Line 337, where does the ">25%" come from? Between which two model runs?

11. Line 378-379, the "14%" refers to the area difference between base and GIA model runs, while the "4.2%" (better change to 4% to be consistent) refers to the thickness difference between base and legacy model runs? This sentence "Our new ... without GIA" is not clear, please rephrase. Similarly, in line 381 "by 14% area and 8% thickness", does it refer to one model or two models.

12. Line 381, delete "report", since AR6 is "sixth Assessment Report".

13, Line 423, remove "Shared Socioeconomic Pathway 8.5 ()", abbreviated in the above context already.

14, Line 466, 480, change "100 a" and "10 a" to "100 yr" and "10 yr", respectively. Units need to be consistent. Both "timesteps" and "time steps" are used (e.g., line 456, 466, & 468, 480), needs to be unified.

15, Line 402, "187m below and 18 m above sea level" is used, while in line 487 "-138 and 18 m asl" is used. Either using only positive numbers with units of m bsl & m asl, or only using unit of m asl with both negative (indicative of below sea level) and positive numbers.

16, Line 488-489, why ruled out locations in big rivers deltas and estuaries? Is that because you

made no assumptions about the locations of paleo rivers and estuaries as stated in line 416-417? Better to make it clearer.

17, Line 506-507, "only permafrost ... a a data quality of a or b ...", not clear what this means.

18, Line 511-512, change "models produce" to "model predictions", add a "differences" after "than".

References

Austermann, J., Mitrovica, J.X., Latychev, K. and Milne, G.A., 2013. Barbados-based estimate of ice volume at Last Glacial Maximum affected by subducted plate. *Nature Geoscience*, 6(7), pp.553-557.

Kuchar, J., Milne, G. and Latychev, K., 2019. The importance of lateral Earth structure for North American glacial isostatic adjustment. *Earth and Planetary Science Letters*, 512, pp.236-245.

Melini, D. and Spada, G., 2019. Some remarks on Glacial Isostatic Adjustment modelling uncertainties. *Geophysical Journal International*, 218(1), pp.401-413.

Patton, H., Andreassen, K., Bjarnadóttir, L.R., Dowdeswell, J.A., Winsborrow, M.C., Noormets, R., Polyak, L., Auriac, A. and Hubbard, A., 2015. Geophysical constraints on the dynamics and retreat of the Barents Sea ice sheet as a paleobenchmark for models of marine ice sheet deglaciation. *Reviews of Geophysics*, 53(4), pp.1051-1098.

Patton, H., Hubbard, A., Andreassen, K., Auriac, A., Whitehouse, P.L., Stroeven, A.P., Shackleton, C., Winsborrow, M., Heyman, J. and Hall, A.M., 2017. Deglaciation of the Eurasian ice sheet complex. *Quaternary Science Reviews*, 169, pp.148-172.

Reviewer #3 (Remarks to the Author):

General

The authors describe a sensitivity analysis where they test the impact of glacial isostatic adjustment (and different past sea level reconstructions) on the modelled extent and thickness of subsea permafrost across the Arctic Ocean. Considering glacial isostatic adjustment overall reduced subsea permafrost thickness, with high spatial variability. Future projections suggest that the fate of subsea permafrost strongly depends on emission scenario, and that high (but not low) emission scenarios will cause the crossing of a tipping point, and dramatic amplification of subsea permafrost thaw.

I overall find the manuscript well written and well balanced, and I have only one main comment: It is difficult for a non-expert on past sea level (like me) to follow the differences between model runs and keep track of the abbreviations. I think some polishing with that in mind would make the manuscript more comprehensible for a broader readership (e.g. scientists interested in Arctic Ocean greenhouse gas emissions).

Specific comments

Lines 38-41: I suggest to re-phrase this sentence to emphasize the large uncertainties, as it otherwise pre-empties your sentence at the end of the paragraph on the need for estimating the amount of carbon in subsea permafrost.

Lines 97-107: I have to say that I am struggling to understand the differences between model runs, also with the many abbreviations. Since these differences are central to your study, I recommend to re-cast this part to make the differences clearer.

Line 139: I suggest to change to "modelled present-day subsea permafrost" or similar

Line 159: Is there a way to say if the base or the legacy run is supposed to be more realistic (based on the input projections)? Or are they to be seen as two options and we do not know at that point which one is more realistic? See also my comment above on differences between runs.

Lines 328-341: This paragraph made me wonder ... on the one hand, less subsea permafrost implies less carbon storage, and less potential for greenhouse gas production by organic matter decomposition after thaw. At the same time, subsea permafrost might also be associated with pre-formed methane within or below (in thermogenic deposits, gas hydrates or simply in gaseous form within the frozen sediment). Subsea permafrost thaw could lead to flow paths for this gas to reach the surface, over smaller or larger areas. Could thinner subsea permafrost as projected by your model increase these risks?

We thank the editor for their diligence in soliciting these reviews and for their careful attention to our manuscript. We respond to the reviewers' comments below, in blue, along with line numbers. We attach a manuscript file with changes tracked, and a cover letter summarizing the main changes to the manuscript. We have also revised the manuscript to comply with the Nature Communications editorial requests and formatting instructions enumerated above. We hope that these changes will be deemed to be satisfactory and look forward to further communications about next steps.

Best wishes,

Roger Creel, on behalf of the coauthors.

REVIEWER COMMENTS

Reviewer #1 (Remarks to the Author):

Review of paper NCOMMS-23-09654: “ Glacial isostatic adjustment reduces past and future arctic subsea permafrost”

I would like to say to the authors that I very much enjoyed reading this paper. It is well written. I suggest it is accepted for publication with some minor changes I list below. They are mostly related to minor changes to terminology and additional information about the method section.

We thank the reviewer for their kind words and appreciate the encouragement. We enjoyed writing the paper also!

Line 133 “Permafrost that exceeds a thickness of 500m also underlies the shallow central Kara Sea..” Should this not be subsea permafrost, as in the rest of the paragraph.

Thank you for this note. We have changed the text as suggested.

Section 2.1.1

When comparing the legacy and base model runs - two factors have been changed: ice sheet history (CLIMBER vs ICE5G) and GMSL curve (Grant and Waelbroek).

To entirely isolate the impact of just GMSL, did the authors consider using the CLIMBER model and Waelbrook curve. As the set-up stands, I am not sure how to separate the ice sheet history and GMSL. Line 157 “ Overall using the base ice history decreases the area of the seafloor presently under lain by permafrost..... “ Is this the choice of ice history or the GMSL curve?

We thank the reviewer for this suggestion, which highlights a point that needs clarifying. Because neither the *legacy* to *base* runs include GIA, variations in ice sheet extent in those runs affect only the permafrost within the footprints of the ice sheets. Any subsea permafrost beyond

the maximum extents of the ice sheets is governed exclusively by GMSL variations, which control the balance of subaerial exposure and flooding. For this reason, it is in fact straightforward to separate the influence of ice sheet history and GMSL for most of the Arctic continental shelf for runs that do not consider GIA. We have added text to section 2.1.1 (lines 195 – 205) that explains this point:

Outside of the maximum extents of the Northern Hemisphere ice sheets, differences in forcing temperature between *legacy* to *base* are entirely explained by the differences in GMSL. Within the footprints of the Northern Hemisphere ice sheets, the differences in forcing temperature are explained principally by differences in ice extent. ...

Line 167 “The change from the legacy to base run diminishes the temperature forcing....” Why does this decrease? Can you expand on why the temperature is cooler in the base run? I have read through the Methods and the description of the temperature calculation is very brief. For example line 420: “Reduction in ice cover extent and duration are expected to warm the sea bed”. Therefore, if the base run produces a cooler temperature is this due to a reduction in ice cover, increased water depth?

This question that the reviewer raises is an important one and we are grateful for the chance to address it. Water depth has a minimal effect in the model. Rather, the critical variable is the amount of time that each grid cell spends covered by water, ice, or air. Any time spent under either ice or water results in subsea permafrost thinning; any time spent exposed to air results in subsea permafrost thickening. We have added the text below to section 2.1.1 (lines 195 – 205) to clarify this point.

Outside of the maximum extents of the Northern Hemisphere ice sheets, differences in forcing temperature between legacy to base are entirely explained by the differences in GMSL. Relative to the legacy run, GMSL in the base run covers shallow continental margin sediments for more time but deep sediments for less time (Fig. 3). Sediments on the shallow continental margin are therefore exposed to warmer ocean temperatures for longer, which inhibits permafrost formation, while deep sediments are exposed to seawater temperatures for less time. Within the footprints of the Northern Hemisphere ice sheets, the differences in forcing temperature are explained principally by differences in ice extent. Any place that is covered by ice for longer during the *base* run than during the *legacy* run has thicker present-day permafrost because basal ice temperatures are warmer than air temperatures; any place with less ice coverage has thinner permafrost.

The reviewer’s question likely arose in part because of the text they highlight in line 420 of the original manuscript, which was misleading. That sentence should (and now does, on line 524) read “Reductions in **sea** ice cover extent and duration are expected to warm the sea bed...”.

Choice of Greenland ice sheet: Both the CLIMBER and ICE6G Greenland ice sheet reconstructions are ‘small’ (as on Fig.5A) and do not really extend beyond the present-day

margin. Would a large Greenland ice sheet impact on your results? - as the continental shelf around Greenland would have a fluctuating ice sheet extent?

We appreciate the reviewer’s question about permafrost around the Greenland ice sheet. We agree that ICE6G is ‘small’, in that the Last Glacial Maximum (LGM) margin does not extend to the edge of the continental shelf. However, we disagree that CLIMBER2 ice sheet is ‘small’. As shown in Fig. 5 from Ganopolski et al. (2010, see Fig. R1 below), the CLIMBER2 ice sheet at 21 kyr BP is substantially more extensive in both East and West Greenland when compared to ICE-5G (Fig R1b), whose Greenland Ice Sheet is identical to the ICE-6G Greenland Ice Sheet. This difference is evident in our Fig. 2B and 2C, which show that while the *legacy* run has no permafrost mapped for the islands along the Greenland coastline (2B), the *base* run maps permafrost of between 100 and 250 m on several small Greenland coastal islands.

Fig. R1. Spatial extent and elevation of **(a)** CLIMBER2 and **(b)** ICE-5G (Peltier, 2004) ice sheets at 21 kyr BP. Reproduced from Ganopolski et al. (2010). Black lines denote shorelines at 21 kyr BP in each model. Purple boxes highlight regions of Greenland Ice Sheet where (a) is smaller than (b).

We have modified section 2.1.1 to note this difference (Lines 189 – 192):

... Areas where *base* run mean temperature forcing is warmer than the *legacy* run, and subsea permafrost consequently thinner, including the Laptev Sea, islands off the coast of West and East Greenland, and the shallower parts of the East Siberian and Chukchi Seas...

As our existing model runs explore this question of model sensitivity, we do not believe that an additional full model run to test a larger Greenland ice sheet is necessary.

Section 2.1.2:

Line 181 “ Gravitational effects tend to be smaller since the rebounding earth in part counteracts the gravitational effects...”

I do not quite follow this sentence. Do you mean the gravitational effects are smaller than the deformational effects, driven by the rebound (shown in Fig. S1b)

The reviewer is justified in not following this sentence, as it was awkwardly worded. We have modified it to read (line 210):

Overall gravitational effects tend to be smaller since the positive gravitational effects of the rebounding Earth in part counteracts the negative gravitational effects from melting ice sheets (Supplemental Fig. S1).

Line 189 “Outboard of the peripheral bulge... which depresses the oceanic crust...” This is an odd terminology, do you mean outside? Beyond? I think it would be more general to say the sea floor, or ocean bed, as it is not just the crust which is depressed.

This is a good point: the syntax was unclear. We have revised the sentence following the reviewer’s suggestion to read (line 220):

“ Outside of the peripheral bulge ... which depresses the sea floor ...”

Line 202: “Inboard of the Laurentide and Eurasian peripheral bulge”. . Again Inboard is odd terminology, Inside, Close to?

We agree that this syntax is ungainly. We have simplified the sentence following the reviewer’s suggestion to read:

“Close to the Laurentide and Eurasian ice sheets, ...”

Line 207 : Reference to Fig. 2A - I think should be Fig.2D and E.

We have changed the reference to Fig. 2E.

Section 2.2:

Great to see the future projections continued to 2300 and 3000 yr.

Fig. 6. Nice investigation of the relation between 1850 thickness and time of disappearance. However, the dark blue dots which make up most of the figure - I am not sure if these relate to high emission (pale purple) or low emission (pale blue)

We agree that the transparency of the dots in this figure make its interpretation harder. We have modified the figure to make the dots more opaque.

Line 242: “GIA affects future... (1) 400,000 years of GIA influence leads to thinner present-day subsea permafrost” How are you assessing that this us 400,000 years of GIA and not 2 glacial cycles or 1 glacial cycles?

We thank the reviewer for pointing out this detail. Overduin et al. 2019 conducted sensitivity tests that showed that present-day subsea permafrost is most sensitive to forcing over the last glacial cycle, somewhat sensitive to forcing during the penultimate glacial cycle, and largely insensitive to forcing during cycles three and four. However, the aim of this sentence was not to highlight the exact number of years over which GIA is important but simply to note that influences of GIA prior to the present control the amount of permafrost in the present. We therefore have revised the sentence to read:

“GIA influence over the late Pleistocene and Holocene leads to thinner ...”

Line 245: “causes local sea level to differ from the mean” revise to “differ from GMSL, to tie in with the rest of the paper terminology.

Noted. We have revised the sentence to read:

“... causes local sea level to differ from GMSL.”

Section 3:

Line 270: “We demonstrate that the ice sheets also influence.... kilometres distant due to the gravitational”

Please change this wording - kilometres distant due to, does not make sense.

We have revised the sentence to read:

“... influence subsea permafrost hundreds to thousands of kilometers beyond their margins because of the gravitational ...”

Line 282:”harnessing subsea permafrost as an ice sheet constraint” harnessing is an odd terminology, please revise .

We agree that this syntax is awkward. We have revised the sentence to read:

“... first steps towards using subsea permafrost to constrain ice sheet histories are already possible ...”

Fig. 8 - nice figure, but can you label what the black dots are?

We have added text in the caption to describe what the black dots are:

“Black dots represent values outside the interquartile range.”

Line 296: “However, modelled Beaufort Sea permafrost in the GIA run is still significantly thinner may be influenced by processes not accounted for in our model”. Rather than this misfit been related to a missing process - why could it not be due to problems with the choice of ice sheet reconstruction. CLIMBER model is not constrained to observational data and ICE6G is just one possible reconstruction. On Fig. 2 the extent of permafrost in the Beaufort Sea is very small or minimal (in the region of 30W to 90W). Are the authors referring to 120W- 170W?

We agree with the reviewer that choice of ice model would also influence the amount of subsea permafrost in the Beaufort Sea region. We note in section 2.1.1 that choice of ice history changes present-day permafrost thickness in the Canadian arctic nearly everywhere by more than 50 m. We have modified this paragraph to read (paragraph starting line 347):

“However, modeled Beaufort Sea permafrost in the *GIA* run is still significantly thinner and shallower than observations. This mismatch suggests that one or more of the forcings used in this study---including ice sheet geometries, relative sea level, and air temperature---are imperfectly representing the region's late Quaternary history; or that subsea permafrost in this region may be influenced by processes not accounted for in our model.”

Fig 9: nice figure.

Thank you!

Line 366: “hastens the sea ice loss” hastens is not a correct word. I would use increases the sea loss or some other phrase.

We have modified this sentence to read “... wherein lost sea ice lowers Arctic albedo, which increases sea ice loss...”

SOM. Fig S3 - please reference in the main paper.

We have modified the text such that all supplemental figures now are mentioned in the main text.

Methods section:

The description of each model, section 5.1 and 5.2 was fine. However, can you expand on how the two models are linked? On line 398 “changed the synthesized forcing temperature by using different sources for sea level, ice sheet histories a...” So, what variable/output from the GIA model is included in the permafrost model in the GIA-runs. Do you use a prediction of RSL? Paleotopography?

We agree with the reviewer that this is an important point for readers to understand. We have included a sentence to clarify this (Lines 521 and following):

“The resulting calculation yields a spatiotemporally-continuous estimation of RSL that is linearly interpolated to the grid centers of the EASE Grid 2.0 and combined with the IBCAO 4.0 bathymetry (see Section 5.1).”

Line 162 “ Mean Surface forcing temperature were calculated at each location from the local history of sea-level, glacial load and air temperature “

Can you clarify this in the method section. Depending on the choice of model (base, legacy or GIA)I assume the ‘local history of sea level’ is either GMSL of Grant, Waelbrook or RSL from GIA model. What is glacial load? Do you mean ice sheet extent or the deformation signal from the GIA model? How is this incorporated in the “non” GIA run.

We thank the reviewer for pointing out this linguistic imprecision. We have replaced ‘glacial load’ with ‘ice sheet extent’ in the relevant sentence.

I think this would be clearer if in the Method section you defined clearly what was the input to the permafrost model and where it was from sourced from, perhaps in a table,

We agree that this presentation would be clearer. Table S1 now includes all forcings including air and seabed temperature.

How do you regrid the GIA model output to the higher resolution of the permafrost model (line 400 12.5km)? On line 466 “interpolated using nearest neighbour interpolation to the 100a timestep of the permafrost simulation” , the authors define how the account for the differences in temporal resolution but not the spatial resolution.

We thank the reviewer for noting this detail. We modify the sentence mentioned to read:

“... timesteps of the permafrost simulation and linear interpolation to the 12.5 km spatial resolution of the EASE Grid 2.0”

Line 482” Model output included sediment temperature and composition’.... A main output that

is evaluated in the paper is thickness, so I would list this as a main output. Sediment temperature and composition are not evaluated, or discussed in the paper.

This point is well taken. We have edited the sentence in question to read:

“Model output included subsea permafrost thickness and ice content ... “

“defined modern elevation”

Line 110 “permafrost was modelled between 136m below and 18m above present day sea level”

Line 402 “included any locations with modern elevations between 187m and 18m above sea level”

Line 487” all locations on the EASE Grid. with current elevations between -138m and 18m asl”

Can you check these numbers and definitions to be consistent? For example, ‘modern’ or ‘current’. I am assuming the 136m is the GMSL maximum, but the RSL variations will be larger? What was the reason for + 18m, +187m?

We thank the reviewer for noting this inconsistency. The correct range is 187 m below to 18 m above sea level, which encompasses the full range of RSL variations, rather than just GMSL variations.

Figures:

Fig.1: “GMSL from Waelbroeck et al., 2002 and Peliter et al., 2015”. I am assuming that the GMSL curve plotted is from the updated ice history described in Section 5.2, as Peltier et al., 2015 does include 400ka results. From reading the caption, I was looking for multiple lines.

We appreciate the reviewer pointing out this syntactic inaccuracy. We have changed the text to read, “Timeseries of subsea permafrost thickness and global mean sea level (GMSL). (A) GMSL from Waelbroeck et al. (2002) (400-26 kyr) and Peltier et al. (2015) (26 ka to 1850 CE, see Methods)”

Fig. 2: Can you add ‘A-GIA run’, ‘B -legacy run’ and ‘C- base run’ to the figure itself. To make the difference plot clearer and to link with the caption “> 200m difference in the permafrost thickness are locations where additional permafrost is introduced” could you add a contour to mark of the edge of the permafrost extent on D and E from the legacy run. It would make it easier to identify where there are new areas of permafrost are simulated in the base and GIA runs.

We agree with the reviewer that these changes would make Fig. 2 more legible and have modified the figure as recommended.

Fig.4: Can you reverse the colour scale. The colder temperature red and warmer temperatures red is counter intuitive.

We appreciate and share the reviewer's desire for intuitive color bars. We have made the change as suggested.

Fig 5a: It is not possible to identify the "ice regions", with the blue colour scale, which looks the same as the negative elevation regions. Can you change the ice covered areas to be different colour shade, and add a scale bar.

We agree with the reviewer that it is important for ice regions to be legible. We have added a scale bar, as suggested. However, having experimented with a variety of other color maps, we find that the original blue map provides the best balance of contrast between the continental margin (light yellow), continent (gray), and benthos (black). We also note that a blue colormap to represent ice is more legible than another map such as green, which is not typically associated with ice. We therefore prefer to keep the blue colormap as is.

Fig 5b and S1: With the colour scale chosen it is not possible to identify the differences between -8m or -4m. Please modify. Can you choose a more contrasting colour? Can you add a contour line for Fig. 5b to mark the edge of the ice sheet regions? As the text describe the regions of peripheral bulge at the edge of the EIS and Laurentide, this will help the reader to identify the edge of the ice sheet. Change exemplary (which refers to something been excellent) to example, or reference.

We appreciate the reviewer's suggestions for making these figures more readable. We have modified the colormaps to contrasting colors, changed them to filled contour plots, and added a zero contour for the ice sheet regions, and replaced 'exemplary' with 'example.'

Can you add a simple table in the SOM of "Model reference name (Legacy, base), ice sheet history, GMSL curve, with/without GIA".

We have included a table following the reviewer's suggestion.

Reviewer #2 (Remarks to the Author):

Review of the manuscript “Glacial Isostatic Adjustment reduces past and future Arctic subsea permafrost” by Creel et al.

The manuscript incorporates the influences of glacial isostatic adjustment (GIA) in the Pan-Arctic model of subsea permafrost for the last 400,000 years and extends the simulation 1000 years into the future for different SSP scenarios. They find that the incorporation of GIA can reduce the present-day subsea permafrost thickness and the subsea permafrost is preserved under a low emissions scenario but mostly disappears under a high emissions scenario.

The paper is generally adequately presented and well organized. The figures are of good quality. However, the GIA doesn't consider the 3D Earth structure, which has been shown having significant impacts on sea-level predictions, especially near the ice-covered regions like Arctic (Austermann et al., 2013; Kuchar et al., 2019), neither the uncertainty related to ice model is considered (e.g., Melini & Spada, 2019). For example, in Fig. 2, the difference between base and legacy model runs is larger than the difference between base and GIA models runs, indicating use of different ice history and GMSL has larger influence (than the incorporation or GIA or not) on the subsea permafrost thickness results at 1850. Moreover, the ice history uncertainty is dramatic in the Arctic region (e.g., Patton et al., 2015, 2017), which needs to be investigated. The authors state the GIA influence on subsea permafrost distribution and state is significant (14%) in the abstract, line 176 and conclusion, while I wonder if the uncertainty is considered, whether the GIA influence is still statistically significant and by how.

We thank the reviewer for their helpful feedback. They are right that 3D Earth structure can have significant impacts on sea-level predictions. However, there are several reasons why the 1D GIA approximation is sufficient for this study:

1. Uncertainties in Arctic RSL introduced by variations in 1D viscosity can be as large as uncertainties associated with differences between 1D and 3D viscosity estimates (Li et al. 2022).
2. Predictions of Arctic RSL using 3D Earth structures are in many regions not able fit Arctic RSL data better than RSL predicted using 1D Earth structures (e.g. Milne et al. 2018). This mismatch persists because many standard ice models, including the ice model highlighted in this study, are optimized to fit RSL data using a 1D viscosity structure; there does not presently exist an ice sheet model optimized to fit RSL data using a 3D viscosity structure.
3. Most importantly, it is not presently possible to perform a 3D GIA calculation over four glacial cycles. To our knowledge, the longest 3D GIA runs that have been published are found in Austermann et al. (2021). These runs, which extended back to 150 ka, took several weeks of parallelized supercomputing cluster time. A 3D GIA run that spanned four glacial cycles would take several months, which is computationally prohibitive.

Nevertheless, we agree that the claim that the GIA influence on subsea permafrost distribution and state would be strengthened with a demonstration of the impact of a differing viscosity structure on present-day permafrost distribution. To that end, we have rerun the permafrost model using ICE-6G with the 1D viscosity structure that Lambeck et al. (2014) found to optimally fit a global RSL dataset. We have included description of this run in the Methods & Supplement section (Lines 525-527 and Fig. S2) and described the results of this sensitivity test on lines 246 – 258:

The amount that GIA reduces subsea permafrost formation depends on solid Earth structure. The *GIA-2* run is identical to the *GIA* run save that the RSL forcing was produced by a GIA model with an overall higher viscosity than the *GIA* run. This viscosity difference increases overall inundation time, particularly in shallow shelf areas, resulting in >50 m less permafrost in shallow continental shelf areas, 10 m to 50 m less permafrost in deeper areas, and <10 m more permafrost in the deepest shelf areas (Supplemental Fig. S2A). The size of the Northern Hemisphere ice sheets also leads to a GIA effect that reduces subsea permafrost formation. The *GIA-3* run is identical to the *GIA-2* run save that during MIS-11 to -7 and MIS-5d to -1 (Fig. 4) the *GIA-3* run uses ANU ice geometries, which include a larger Eurasian ice sheet than ICE-6G does. This difference increases the Eurasian Ice Sheet's gravitational and deformational influence on Arctic sea level, leading to 10 m to 50 m less permafrost across much of the Arctic continental shelf (difference between Supplemental Fig. S2A and S2B).

We also agree that uncertainty associated with the choice of ice sheet history is important to consider. Changing the ice history has several effects: (1) It changes the GMSL history, (2) it changes the footprint of the ice sheet (and consequently the permafrost beneath the ice), and (3) it changes RSL due to GIA. We now clarified language in section 2.1.1 detailing effects (1) and (2) (lines 157 – 163, additions bolded):

GMSL in the *base* run is generally higher early in glacial intervals (MIS 11b-10b, 9d-8b, 7b-6b, 5d-3a) than GMSL in the *legacy* run, but lower during peak glacials (MIS 10a, 7d, 6a, 2, Fig. 3). **This difference in GMSL has a pan-Arctic effect on subsea permafrost.** Higher early-glacial GMSL inhibits the formation of shallow subsea permafrost everywhere in the Arctic by decreasing subaerial exposure time; lower peak-glacial GMSL enhances subsea permafrost formation on the deep shelf (Fig. 3). **Subsea permafrost differences driven by ice sheet geometry are limited in extent to areas covered by grounded ice.** For instance, the >500m thickness difference in the eastern Kara Sea is caused by differences in ice distribution.

In our revisions, we have taken an additional step to assess uncertainties associated with effects (2) and (3) above. We have produced an ice history that follows the same GMSL history as the GMSL used in the *GIA* run but has contrasting ice geometries. The ice history is produced by pairing the northern hemisphere ice sheets from Lambeck and colleagues (1995; 2010, 2014; 2017) with an Antarctic ice history based off of ICE-6G but scaled in order that the sum of ice sheet contributions to GMSL equals the GMSL of the *GIA* run (see Methods lines 543 – 552). We then predict relative sea level from the new ice model using the viscosity preferred by

Lambeck and colleagues (2014), and rerun the permafrost model with the new sea level and ice sheet forcing. The results, described on lines 246 - 258, demonstrate that though GIA does vary with ice sheet geometry, the fact that present-day subsea permafrost is reduced by GIA remains and is not an artifact of the chosen ice history. Rather, the sensitivity tests confirm that our preferred model is, if anything, a conservative estimate – conservative in the sense of our preferred model reducing permafrost by less than the sensitivity runs – which bolsters our main argument.

We highlight that the permafrost model is computationally expensive and takes more than a month for one (400kyr) simulation. This limits us to a small set of sensitivity tests and we hope that the ones we have chosen here address the concerns that the reviewer raises.

Detailed comments:

1, Line 18, “International Panel on Climate Change’s sixth assessment report” should be changed to “Intergovernmental Panel on Climate Change sixth Assessment Report”. Same in line 94.

We have made this change as suggested.

2. Line 101, change “(Waelbroeck et al., 2002)” to “Waelbroeck et al. (2002)”

We have made this change as suggested.

3. In Fig. 2, label the locations. Although you commented “see Fig. 5A for locations”, it is not readers-friendly to go back and forward and the locations are already referred in line 132-135. And in the fig caption, the last sentence “Areas ...” is not clear to me, does it mean the areas with >200 m difference are the places that in legacy run have no subsea permafrost? If yes, better to distinguish with different color, as based on the legend, the colors are mainly to show the depth difference, can use other color/map to indicate the subsea permafrost geographical coverage difference between different model runs.

We have added labels as suggested and modified the caption of Fig. 2 to read, ‘Areas in (D)/(E) with >200 m difference in permafrost thickness are locations where no permafrost is present in the legacy/base case but permafrost is introduced in the base/GIA cases, respectively.’

4. Line 142, the “>500 m” cannot be identified in the plot. Change “fig. 2B” to “Fig. 2D”.

We have modified this text to better align with the colormaps and changed the figure number as suggested.

5. Line 171, change “2B)” to “(Fig. 2D)”. And in line 173, change “include” to “including”.

We have made the changes as suggested.

6. Line 178, & 217-218, how about the differences between legacy and base runs? Both for the subsea permafrost coverage area and mean thickness.

We agree that this is an important difference to note and point the reviewer to line 158, where we describe the differences between legacy and base runs in language similar to the language in 217-218.

7. Fig. 6 caption, “loss percentage and thickness”, does the thickness refers to the remaining permafrost thickness? If yes, may add a “remaining” before “thickness”.

We have made the changes as suggested.

8. Line 260-261, 2260 vs 2290, shouldn't be the difference ~30 years? Please label 2260 and 2290 in Fig 7 (maybe a vertical line to mark the year).

We have made the changes as suggested and added text to the caption to describe the change:

“Dashed lines represent time when all permafrost thinner than 100 m has disappeared.”

9. Line 332-333, both “gigatons” and “Gt” are used, needs to be unified.

We have unified the units as suggested.

10. Line 337, where does the “>25%” come from? Between which two model runs?

We appreciate this request for more clarity and have modified the text to read: “We also find that the area of seafloor presently underlain by subsea permafrost, as estimated in the GIA run, is >25 % smaller than previously estimated (Overduin et al., 2019)”

11. Line 378-379, the “14%” refers to the area difference between base and GIA model runs, while the “4.2%” (better change to 4% to be consistent) refers to the thickness difference between base and legacy model runs? This sentence “Our new ... without GIA” is not clear, please rephrase. Similarly, in line 381 “by 14% area and 8% thickness”, does it refer to one

model or two models.

In response to the reviewer's fair point, we have reworded this paragraph:

Our new subsea permafrost map, based on the GIA run, has 14 % less seafloor area underlain by permafrost and is 4 % thinner than the base run. Both the GIA and base runs update the ice cover and sea level forcing of the legacy run (cf. P. P. Overduin et al., 2019), resulting in even less permafrost: the base run has 14 % less area and is 8 % thinner than the legacy run.

12. Line 381, delete "report", since AR6 is "sixth Assessment Report".

We have made the change as suggested.

13, Line 423, remove "Shared Socioeconomic Pathway 8.5 ()", abbreviated in the above context already.

We have made the change as suggested.

14, Line 466, 480, change "100 a" and "10 a" to "100 yr" and "10 yr", respectively. Units need to be consistent. Both "timesteps" and "time steps" are used (e.g., line 456, 466, & 468, 480), needs to be unified.

We have made the change as suggested.

15, Line 402, "187m below and 18 m above sea level" is used, while in line 487 "-138 and 18 m asl" is used. Either using only positive numbers with units of m bsl & m asl, or only using unit of m asl with both negative (indicative of below sea level) and positive numbers.

We have made the change as suggested (see above).

16, Line 488-489, why ruled out locations in big rivers deltas and estuaries? Is that because you made no assumptions about the locations of paleo rivers and estuaries as stated in line 416-417? Better to make it clearer.

We thank the reviewer for pointing out this omitted explanation. We have added a line to explain the choice:

This filter is applied because the permafrost in those locations is likely mis-estimated due to its dependence on estuarine sedimentary processes which are not represented in our model.

17, Line 506-507, “only permafrost ... a a data quality of a or b ...”, not clear what this means.

We appreciate the request for clarity and have changed the text to read:

“only permafrost lower limit observations of high data quality (i.e. the a or b categories from Hu et al. (2013)) were included from Hu et al. (2013)”

18, Line 511-512, change “models produce” to “model predictions”, add a “differences” after “than”.

We have made the change as suggested.

References

Austermann, J., Mitrovica, J.X., Latychev, K. and Milne, G.A., 2013. Barbados-based estimate of ice volume at Last Glacial Maximum affected by subducted plate. *Nature Geoscience*, 6(7), pp.553-557.

Kuchar, J., Milne, G. and Latychev, K., 2019. The importance of lateral Earth structure for North American glacial isostatic adjustment. *Earth and Planetary Science Letters*, 512, pp.236-245.

Melini, D. and Spada, G., 2019. Some remarks on Glacial Isostatic Adjustment modelling uncertainties. *Geophysical Journal International*, 218(1), pp.401-413.

Patton, H., Andreassen, K., Bjarnadóttir, L.R., Dowdeswell, J.A., Winsborrow, M.C., Noormets, R., Polyak, L., Auriac, A. and Hubbard, A., 2015. Geophysical constraints on the dynamics and retreat of the Barents Sea ice sheet as a paleobenchmark for models of marine ice sheet deglaciation. *Reviews of Geophysics*, 53(4), pp.1051-1098.

Patton, H., Hubbard, A., Andreassen, K., Auriac, A., Whitehouse, P.L., Stroeven, A.P., Shackleton, C., Winsborrow, M., Heyman, J. and Hall, A.M., 2017. Deglaciation of the Eurasian ice sheet complex. *Quaternary Science Reviews*, 169, pp.148-172.

Reviewer #3 (Remarks to the Author):

General

The authors describe a sensitivity analysis where they test the impact of glacial isostatic adjustment (and different past sea level reconstructions) on the modelled extent and thickness of subsea permafrost across the Arctic Ocean. Considering glacial isostatic adjustment overall reduced subsea permafrost thickness, with high spatial variability. Future projections suggest that the fate of subsea permafrost strongly depends on emission scenario, and that high (but not low) emission scenarios will cause the crossing of a tipping point, and dramatic amplification of subsea permafrost thaw.

I overall find the manuscript well written and well balanced, and I have only one main comment: It is difficult for a non-expert on past sea level (like me) to follow the differences between model runs and keep track of the abbreviations. I think some polishing with that in mind would make the manuscript more comprehensible for a broader readership (e.g. scientists interested in Arctic Ocean greenhouse gas emissions).

We are grateful to the reviewer for their positive feedback and agree that the model runs are challenging to keep straight. To address this point, which reviewer 1 also made, we have added a table in the supplementary materials that lists the salient characteristics of each run.

Specific comments

Lines 38-41: I suggest to re-phrase this sentence to emphasize the large uncertainties, as it otherwise pre-empties your sentence at the end of the paragraph on the need for estimating the amount of carbon in subsea permafrost.

This is a good point, and we thank the reviewer for flagging it. We have modified the cited text to read:

Recent work and structured expert assessment, however, suggest that the submarine permafrost domain may hold an amount of carbon in organic matter and methane hydrates of similar magnitude to the Earth's total gas reserves ... Since this carbon may reach the atmosphere as greenhouse gas, it is important to have a more precise estimate for the amount ...

Lines 97-107: I have to say that I am struggling to understand the differences between model runs, also with the many abbreviations. Since these differences are central to your study, I recommend to re-cast this part to make the differences clearer.

We agree that the original manuscript was unclear in its differentiation between model runs. We have constructed a supplementary table (Table S1) to clarify the models. We have also modified the manuscript in several places to make the differences more clear (see responses to reviewers 1 and 2).

Line 139: I suggest to change to “modelled present-day subsea permafrost” or similar

We have made the change as suggested.

Line 159: Is there a way to say if the base or the legacy run is supposed to be more realistic (based on the input projections)? Or are they to be seen as two options and we do not know at that point which one is more realistic? See also my comment above on differences between runs.

We thank the reviewer for this suggestion and have changed the relevant text to read:

The larger EIS footprint in the base run better conforms to observational evidence of EIS extent than the legacy EIS (Lambeck et al., 2006; Lambeck, 1995), suggesting that adopting the *base* run ice history may improve the accuracy of the subsea permafrost distribution modeled here.

Lines 328-341: This paragraph made me wonder ... on the one hand, less subsea permafrost implies less carbon storage, and less potential for greenhouse gas production by organic matter decomposition after thaw. At the same time, subsea permafrost might also be associated with pre-formed methane within or below (in thermogenic deposits, gas hydrates or simply in gaseous form within the frozen sediment). Subsea permafrost thaw could lead to flow paths for this gas to reach the surface, over smaller or larger areas. Could thinner subsea permafrost as projected by your model increase these risks?

We thank the reviewer for this insight and agree. We have added this text to account for the possibility that the reviewer notes (line 417 – 419):

The rate at which subsea permafrost may release methane is also largely unknown. If ocean bottom warming can more easily destabilize methane associated with thinner subsea permafrost, our findings may increase the near-term climate risks that subsea permafrost thaw poses.

Reviewer #1 (Remarks to the Author):

Thank you for the in depth responses to my comments. I think your paper should be published following these responses.

Reviewer #2 (Remarks to the Author):

Review of the revised manuscript "Glacial Isostatic Adjustment reduces past and future Arctic subsea permafrost" by Creel et al.

Authors have made sufficient revisions to address my comments and concerns, appropriate responses are provided. The current manuscript is in good shape for publication in NC after some minor edits are made as follow.

Detailed comments:

- 1, Line 146, remove ", see Fig. 3A for locations" as location names are provided in Fig. 2A.
2. Fig. 1 caption, in the first sentence, change "subsea permafrost thickness" to "mean permafrost thickness" to be consistent with the Fig. 1B vertical axis.
3. Line 162, ">500 m"? It is >200 m in line 152, please double check.
4. Fig. 3 caption, "26 ka", kyr is widely used in the text and figures, please change "ka" to "kyr". Similarly in Fig. 4, 5 captions, and in line 546.
5. Line 244, change "present day" to "present-day" to be consistent with other places (e.g., lines 15, 28).
6. Line 273, change "glacial isostatic adjustment" to "GIA".
7. Line 275-277, here says "lower bounds", while in 478-479 sates "upper bound".
8. Line 364, change "relative sea level" to "RSL".
9. Line 388, 391, delete "T. ".

Glacial Isostatic Adjustment reduces past and future subsea permafrost Response to reviewers

1. Line 146, remove “, see Fig. 3A for locations” as location names are provided in Fig. 2A.

Changed, thanks.

2. Fig. 1 caption, in the first sentence, change “subsea permafrost thickness” to “mean permafrost thickness” to be consistent with the Fig. 1B vertical axis.

Changed, thanks.

3. Line 162, “>500 m”? It is >200 m in line 152, please double check.

This is now consistently >200 m.

4. Fig. 3 caption, “26 ka”, kyr is widely used in the text and figures, please change “ka” to “kyr”. Similarly in Fig. 4, 5 captions, and in line 546.

Made consistent, thanks.

5. Line 244, change “present day” to “present-day” to be consistent with other places (e.g., lines 15, 28).

Changed, thanks.

6. Line 273, change “glacial isostatic adjustment” to “GIA”.

Changed, thank you.

7. Line 275-277, here says “lower bounds”, while in 478-479 states “upper bound”.

We have now made this consistent.

8. Line 364, change “relative sea level” to “RSL”.

Changed, thank you.

9. Line 388, 391, delete “T. ”.

Changed, thank you.